

1       **PHENOMENOLOGY OF HIGH OZONE EPISODES IN NE SPAIN**

Xavier QUEROL[1], Gotzon GANGOITI [2], Enrique MANTILLA[3], Andrés ALASTUEY[1], Maria Cruz
MINGUILLÓN[1], Fulvio AMATO[1], Cristina RECHE[1], Mar VIANA[1], Teresa MORENO[1], Angeliki
KARANASIOU[1], Ioar RIVAS[1], Noemí PÉREZ[1], Anna RIPOLL[1], Mariola BRINES[1], Marina EALO[1], Marco
PANDOLFI[1], Hong-Ku LEE[4], Hee-Ram EUN[4], Yong-Hee PARK[4], Miguel ESCUDERO[5], David BEDDOWS[6],
Roy M. HARRISON[6+], Amelie BERTRAND[7], Nicolas MARCHAND[7], Andrei LYASOTA[8],Bernat CODINA[8],
Miriam OLID[8], Mireia UDINA[7], Bernat JIMÉNEZ B.[8], Rosa M. SOLER[8], Lucio ALONSO[2], Millán
MILLÁN [3],Kang-Ho Ahn[4]
[1] Institute of Environmental Assessment and Water Research, IDAEA-CSIC, C/ Jordi Girona 18-26, 08034 Barcelona, Spain
[2] Escuela Técnica Superior Ingeniería de Bilbao, Departamento Ingeniería Química y del Medio Ambiente, Universidad
del País Vasco UPV/EHU, Urkixo Zumarkalea, S/N, 48013 Bilbao, Spain
[3] Centro de Estudios Ambientales del Mediterráneo, CEAM, Unidad Asociada al CSIC, Parque Tecnológico C/ Charles R.
Darwin, 14 46980 Paterna, Valencia, Spain
[4] Department of Mechanical Engineering, Hanyang University, Ansan 425-791, Republic of Korea
[5] Centro Universitario de la Defensa de Zaragoza, Academia General Militar, Ctra. de Huesca s/n, 50090 Zaragoza, Spain
[6] Division of Environmental Health & Risk Management. School of Geography, Earth & Environmental Sciences.
University of Birmingham. Edgbaston, Birmingham B15 2TT, UK
[7] Aix Marseille Université, CNRS, LCE UMR 7376, 13331 Marseille, France
[8] Department of Astronomy and Meteorology, Faculty of Physics, University of Barcelona, Martí I Franquès 1, 08028
Barcelona, Spain
[+]Also at: Department of Environmental Sciences/Center for Excellence in Evironmental Studies, King Abdulaziz
University, Jeddah, Saudi Arabia.
**ABSTRACT**
Ground level and vertical measurements (coupled with modelling) of ozone ($O_3$), other gaseous
pollutants (NO, $NO_2$, CO, $SO_2$) and aerosols were carried out in the plains (Vic Plain) and valleys of
the northern region of the Barcelona Metropolitan Area (BMA) in July 2015; an area typically
recording the highest $O_3$ episodes in Spain. Our results suggest that these very high $O_3$ episodes
were originated by three main contributions: (i) the surface fumigation from high $O_3$ reservoir
layers located at 1500-3000 m a.g.l., and originated during the previous day(s) injections of
polluted air masses at high altitude; (ii) local/regional photochemical production and transport (at
lower heights) from the BMA and the surrounding coastal settlements, into the inland valleys; and
(iii) external (to the study area) contributions of both $O_3$ and precursors. These processes gave rise
to maximal $O_3$ levels in the inland plains and valleys northwards from the BMA when compared to
the higher mountain sites. Thus, a maximum $O_3$ concentration was observed within the lower
tropospheric layer, characterised by an upward increase of $O_3$ and black carbon (BC) up to around
100-200 m a.g.l. (reaching up to 300 μg/m$^3$ of $O_3$ as a 10-s average), followed by a decrease of
both pollutants at higher altitudes, where BC and $O_3$ concentrations alternate in layers with
parallel variations, probably as a consequence of the atmospheric transport from the BMA and the
return flows (to the sea) of strata injected at certain heights the previous day(s). At the highest
altitudes reached in this study (900-1000 m a.g.l.) during the campaign, BC and $O_3$ were often anti-
correlated or unrelated, possibly due to a prevailing regional/hemispheric contribution of $O_3$ at
those altitudes. In the central hours of the days a homogeneous $O_3$ distribution was evidenced for
the lowest 1km of the atmosphere, although probably important variations could be expected at
higher levels, where the high $O_3$ return strata are injected according to the modelling results and
free sounding data.





Relatively low concentrations of ultrafine particles (UFP) were recorded in the 100-200 m a.g.l.
atmospheric layer where concentrations of $O_3$ were high; and nucleation episodes were only
detected into the boundary layer.
Two types of $O_3$ episodes were identified: Type A) with major exceedances of the $O_3$ information
threshold (180 $\mu g/m^3$ on an hourly basis) caused by a clear daily concatenation of local/regional
production with accumulation (at upper levels), fumigation and direct transport from the BMA
(closed circulation); and Type B) with regional $O_3$ production without major recirculation (neither
fumigation) of the polluted BMA/regional air masses (open circulation), and relatively lower $O_3$
levels.
The interpretation of $O_x$ ($O_3$+$NO_2$) experimental data from strategically selected monitoring sites
on the coast and inland, together with the photochemical modelling results have allowed to study
the $O_3$ phenomenology associated with the onset and development of severe episodes in the
region of Catalonia in NE Spain.
To implement potential $O_3$ control and abatement strategies two major key tasks are proposed: (i)
meteorological forecasting, from June to August, to predict recirculation episodes so that $NO_x$ and
VOCs abatement measures can be applied before these episodes start; (ii) sensitivity analysis with
high resolution modelling to evaluate the effectiveness of these potential abatement measures of
precursors for $O_3$ reduction.
**Keywords:** $O_3$, photochemistry, air pollution, air quality, $NO_x$

## 68 INTRODUCTION

Ozone ($O_3$) is an airborne secondary pollutant that is produced through the photo oxidation of
volatile organic compounds (VOCs) in the presence of nitrogen oxides ($NO_x$=$NO$+$NO_2$), with more
intensive production in high insolation regions. It is well known that its formation processes are
very complex and that the reaction and production rates are not linear (Monks et al., 2015 and
references therein). According to EEA (2015) 97% of the European population is exposed to $O_3$
concentrations that exceed the WHO guideline (see below) for the protection of the human
health. The complexity of this pollutant is also reflected in its air quality targets; thus, the
European air quality directive 2008/50/EC establishes a number of $O_3$ target values (which are not
legally binding, as opposed to the limit values set for the majority of pollutants):
• A human health target value fixed at 120 $\mu g/m^3$ as 8 hours maxima in a day that should
not be exceeded in more than 25 days/year as a three-year mean. This target value was
(arbitrarily) increased from the recommended 100 $\mu g/m^3$ in the WHO air quality guidelines
(where no exceedances are recommended).
• A population information hourly threshold of 180 $\mu g/m^3$.
• A population alert hourly threshold of 240 $\mu g/m^3$.
• A vegetation protection target, AOT40 [expressed in $\mu g/m^3 \cdot h$], as the sum of the excess of
hourly concentrations above 80 $\mu g/m^3$ along a given period using only hourly values
measured between 8:00 and 20:00 h, Central Europe Time (CET), for every day. Hourly
AOT40 from May to July should not exceed 18.000 $\mu g/m^3 \cdot h$ $O_3$ as a mean for 5 years.
$NO_x$ has a catalytic effect in $O_3$ generation, and is only removed from the system by either
deposition or oxidation to nitric acid ($HNO_3$) and reaction with VOCs to yield secondary aerosols.
Consequently, $O_3$ generation involves not only local and regional air masses but also long-range





transport. Thus, as a general observation, long range transport of $O_3$ and its precursors influence
markedly the background $O_3$ levels in Europe (UNECE, 2010; Doherty et al., 2013). However, this
situation might be very different when considering the high summer $O_3$ episodes of Southern
Europe (e.g. Millán et al., 1997, 2000; Palacios et al., 2002; Castell et al., 2008a, 2008b, 2012; Stein
et al., 2005; Escudero et al., 2014; Pay et al., 2014; Querol et al., 2016).
In the Western Mediterranean basin the problem of tropospheric $O_3$ has been intensively studied
since the early 1980s (Millán et al., 1991, 1996a, 1996b, 1996c, 2000, 2002; Millán, 2002a; Millán
and Sanz, 1999; Mantilla et al., 1997; Salvador et al., 1997, 1999; Gangoiti et al., 2001; Stein et al.,
2004, 2005; Doval et al., 2012; Castell et al., 2008a, 2008b, 2012; Escudero et al., 2014). Results
have evidenced that (i) the meteorology driving $O_3$ fluctuation in this region is markedly influenced
by a very complex orography with high mountain chains surrounding the basin; (ii) in summer, the
lack of a marked synoptic advection caused by the presence of the Azores anticyclone and the
Iberian and north African thermal lows, together with the sea and land breezes, give rise to air
mass recirculation episodes (lasting for several days); and (iii) during these summer vertical and
horizontal recirculations of air masses loaded with $O_3$ precursors and coinciding with high
insolation and elevated biogenic VOCs (BVOCs) emissions (Seco et al., 2011), high $O_3$
concentrations may be recorded.
Millán's team results demonstrated that Western Mediterranean basin dynamics are very
different from those in Central Europe. The latter are dominated by neutral-cloudy conditions,
where $O_3$ episodes are usually associated with advection, and transformation takes place within
large displacements of air masses. In days, morning fumigation from a high $O_3$ residual (and
stratified) boundary layer (BL) formed over the previous days, in addition to local formation in the
sunny midday period may give rise to peak $O_3$ episodes if conditions persist after several days. In
contrast, vertical re-circulations developed over all Western Mediterranean coastal areas,
determine a very different $O_3$ dynamics. Air masses travel all the way from the sea to the
continental divide, or to the top of the Apennines in the case of Italy. These air mass circulations
create layers over the sea at various altitudes, with accumulated pollutants/precursors in several
stages of transformation. These processes can occur during a few consecutive days (e.g. 10 days,
Millán et al., 1997). The layers already over the basin descend from 1000-1500 m a.s.l. during the
day and can reach the lower levels, providing a high background $O_3$ to coastal cities when the sea
and up-slope breezes build up (Millán et al., 2000). Layers, by definition, are stratified and
decoupled from each other so they can move in different directions and speeds at their own
heights.
Rodriguez et al. (2002) and Cusack et al. (2013) showed that these high background $O_3$ episodes
are characterised also by high particulate matter (PM) concentrations, mostly due to the
formation of secondary organic and inorganic aerosols. Such episodes are very common from June
to August, and are usually limited by the occurrence of episodic Atlantic or African advective
conditions that help (especially the first) to ventilate the Western Mediterranean basin. Minguillón
et al. (2015) demonstrated also the occurrence of very intense aerosol nucleation episodes under
high insolation scenarios in the vertical column (from 200 to 1000 m.a.s.l.) over the city of
Barcelona. As the surface air ascends, aerosols are diluted and levels of $O_3$ are expected to
increase.
The Barcelona Metropolitan Area (BMA) is a highly industrialised and dense urban agglomeration
extending over the Mediterranean side of northeast Spain. High anthropogenic $NO_x$ emissions
arise both from road (and shipping) traffic and power generation, which combined with BVOCs
emissions, very often cause severe $O_3$ episodes in the northern plains and valleys (Toll and





Baldasano, 2000; Barros et al., 2003; Gonçalves et al., 2009; Valverde et al., 2016; Querol et al.,
2016). The urban plume is transported inland by sea breezes, heading North channelled by N-S
valleys that cross the coastal and pre-coastal Catalan Ranges to an intra-mountain plain (the Vic
Plain) where the cities of Manlleu, Vic and Tona lie, 40-65 km north of Barcelona (Figure 1). A
mean of 15 annual exceedances of the hourly $O_3$ information threshold/site are recorded at the
urban background monitoring sites of these cities (Querol et al., 2016). In 2015, 96 out of the 115
hours exceeding the $O_3$ information threshold in the whole Catalonia air quality monitoring
network were recorded in the area within 40-90 km north of Barcelona (towards the Pyrenees),
and 82 in the Vic Plain itself (http://www.gencat.cat/mediamb/qaire/ciozo.htm).
This work focuses on an intensive campaign on $O_3$ and particulate pollutants performed in and
around the BMA during July 2015, when high $O_3$ episodes were recorded, with the aim of
investigating the origin of the most intense $O_3$ events in north-eastern Spain. To this end, regional
air quality monitoring network data, passive dosimeters at ground level, vertical profile
measurements of $O_3$ and ultrafine particles (UFP) in the Vic Plain, and modelling tools were
employed.

**METHODOLOGY**
**Study area**
This study is set in central Catalonia (Figure 1), in north-eastern Spain. The mountain ranges
surrounding the area (Pyrenees and Catalan Coastal Ranges) protect the area from the advection
of Atlantic and continental air masses but hamper dispersion of pollutants. The typical winds in the
region are the Tramontana (northern winds), the Mistral or Cierzo (north-western winds
channelled by the Ebro valley) and the sea breezes in the coastal region. In summer, daytime up-
slope winds combined with sea breezes may result in air masses penetrating 120-160 km inland
that are injected aloft the top of the mountains, and follow the return night flows towards the sea
(Millán, 2014). This scenario of air mass regional recirculation during periods of several days
prevails in summer (Millán et al., 1997 and 2000). Hence, summer pollution events are
characterised by (i) the absence of large-scale forcing and the predominance of mesoscale
circulations; (ii) the formation of a thermal low at a peninsular level (forcing the convergence of
surface winds from the coastal areas towards the central plateau with strong levels of subsidence
over the Western Mediterranean basin); and (iii) combined breeze dynamics, resulting in the
recirculation and accumulation of pollutants over the whole Western Mediterranean basin,
including the Eastern Iberian peninsula (Millán 2014 and Millán et al., 1997, 2000).
The region is characterised by important atmospheric pollutant emissions from road traffic,
industries, biomass burning, livestock, and airport and shipping activities, which coupled with high
solar radiation turns into a high rate of secondary PM and $O_3$ formation (Rodriguez et al., 2002).
Industrial activities are mostly concentrated in the Barcelona and Tarragona provinces, and include
19 combustion/energy plants, 84 metallurgy plants and 70 mineral industries. Road traffic, airport
and shipping emissions are concentrated in the Barcelona area with >3.5 $10^6$ vehicles (0.6 per
inhabitant, with high diesel and motorbike proportions; DGT, 2014), >45 $10^6$ tons of shipping
transportation, and >37 $10^6$ aircraft passengers in 2014 (Ajuntament de Barcelona, 2015).
Agriculture, livestock and biomass burning emissions are spread over the rural areas but
concentrated in the core study area, the Vic Plain: a 30 km long depression in the north-south





direction located 60 km to the north of Barcelona. The area is surrounded by mountains is affected
by thermal inversions during the night. The summer atmospheric dynamics dominated by sea
breezes from the southern sector, channelled through the valleys formed by the coastal ranges,
giving rise to the transport of pollutants from the BMA and the numerous surrounding highways.

**Ground level measurements**
*Automatic measurements of gaseous pollutants*
Measurements of gaseous pollutants were performed at 48 sites belonging to the regional air
quality network (Figure 1, XVPCA; http://dtes.gencat.cat/icqa, Table S1) from 01 to 31/07/2015.
Continuous measurements of $O_3$, NO, $NO_2$, CO and $SO_2$ were carried out using MCV 48AV UV
photometry analysers, Thermo Scientific chemiluminescence analysers (42i-TL), Teledyne 300 EU
Gas filter correlation analysers; and Teledyne 100 EU UV fluorescence analysers, respectively.
*Measurements of gaseous pollutants with passive dosimeters*
Diffusion tubes for $NO_2$ and $O_3$ sampling (Gradko Environmental) were deployed at 17 locations
between the cities of Barcelona and Ripoll (Figure 1) covering strategic areas not monitored by the
regional air quality network. The dosimeters were positioned along two main river basins in the
study area (Besòs/Congost and Tordera), from the BMA to the Vic Plain. Sampling points were
selected avoiding the direct influence of vehicular emissions and located at a height of
approximately 2.5 m above ground level. One sample per site and sampling period (01-14 and 14-
29/07/2015) were collected. After exposure, samples were stored at 4 °C until analysis.
Replicas were placed in 9 locations, showing good reproducibility of the results (relative errors of
5%±6% for $O_3$ and 4%±7% for $NO_2$). Dosimeters were collocated also at some XVPCA sites for
comparison with reference measurements (6 samples for $O_3$ at Vic (VIC), Montcada (MON) and
Montseny (MSY); and 10 samples for $NO_2$ at Santa María de Palau Tordera (SMPT), Palau Reial
(PLR), Tona (TON), Manlleu (MAN), and Granollers (GRA)). Correction factors were obtained from
the comparison between dosimeter and reference data (Dosimeter $O_3$ = 1.01*Reference $O_3$ +
17.43, $R^2$ = 0.97; and Dosimeter $NO_2$ = 1.27*Reference $NO_2$ + 1.14, $R^2$ = 0.90, in all cases in $\mu g/m^3$)
and then applied to the dosimeter data (supplementary information Figure S1).
$NH_3$ passive samplers (CEH ALPHA, Tang et al., 2001) were also used in 30 specific points.
*$O_x$ concentrations*
$O_x$ values ($NO_2$+$O_3$) were calculated to complement the interpretation of $O_3$ concentrations. The
concept of $O_X$ was initially proposed by Kley and Geiss (1994) to analyse the $O_3$ spatial and time
variability by diminishing the effect of titration of $O_3$ by NO (NO+$O_3$ → $NO_2$+$O_2$, with the
subsequent consumption of $O_3$) in highly polluted areas with high NO concentrations.
*Laboratory van in Vic*
A laboratory van was deployed next to the Vic air quality station during 10-17/07/2015. Eight-hour
$PM_{2.5}$ samples were collected three times per day (00:00-08:00, 08:00-16:00, and 16:00-00:00
UTC) by means of Digitel DH80 high volume samplers (30 $m^3$/h) on Pallflex quartz fiber filters (QAT
UP). Filters were conditioned at 20-25°C and 50% relative humidity over at least 24 h before
and after sampling to determine gravimetrically the $PM_{2.5}$ concentration. Subsequently a detailed
chemical analysis following the procedure described by Querol et al. (2001) was carried out.



Hourly equivalent Black Carbon (BC) in $PM_{2.5}$ concentrations were determined by a multi-angle
absorption photometer (MAAP, model 5012, Thermo). $PM_1$, $PM_{2.5}$ and $PM_{10}$ hourly concentrations
were determined by an optical particle counter (GRIMM 1107).

**Vertical profiles**

During 14-17/07/2015 several vertical profiles up to 1550 m a.s.l. (1000 m a.g.l.) were performed
by means of a tethered balloon (details can be found in Table S2) in the city of Vic (Figure 1), at
less than 200 m from the laboratory van and the Vic air quality station. The tethered balloon of 27
$m^3$ filled with helium was equipped with an instrumentation pack attached 30 m below the
balloon. This setting has been successfully used in previous studies (Minguillón et al., 2015), hence
the lack of a fixed support for the instrumentation pack is not expected to hinder the quality of
measurements.
The instruments included in the pack were:

- A miniaturized condensation particle counter (Hy-CPC) measuring particle number concentration larger than 3 nm with a time resolution of 1 s and a flow rate of 0.125 L/min, using isopropyl alcohol as the working fluid (Lee et al., 2014). The particle number concentration measured by the Hy-CPC will be referred to as $N_3$. Prior studies have demonstrated the agreement of the Hy-CPC and conventional TSI CPCs (Minguillón et al., 2015).
- A miniaturized nano-particle sizer for the determination of the particle number size distribution (Hy-SMPS, Figure S2) in the range 8-245 nm with a time resolution of 45 s and a flow rate of 0.125 L/min (Lee et al., 2015). The instrument output agreed well with the results from a Scanning Mobility Particle Sizer (SMPS), composed by a Differential Mobility Analyser (DMA, TSI 3081) coupled with a CPC (TSI 3772) (Figure S3, Hy-SMPS= 0. 71*Reference SMPS + 999, $R^2$ = 0.88). SPECIFY SIZE RANGES TO ACCOUNT FOR THIS LARGE DIFFERENCES
- A miniaturized optical particle counter (Hy-OPC, Figure S2) measuring particle number concentrations in the ranges 0.3-0.5 µm ($N_{03-0.5}$), 0.5-1.0 µm ($N_{0.5-1.0}$), 1.0-2.0 µm ($N_{1.0-2.0}$), and 2.0-5.0 µm ($N_{2.0-5.0}$), with a time resolution of 1 s and a flow rate of 1 L/min.
- A microaethalometer (MicroAeth AE51), which provided BC concentrations derived from absorption measurements on a 5 min basis with a flow rate of 0.15 L/min.
- A portable $O_3$ monitor that measures concentrations every 10 s based on UV absorption (POM$^{TM}$ 2B Technologies, Figure S2). The personal POM $O_3$ monitor was compared with the $O_3$ concentrations from the nearby reference station, yielding good results (n=34 min data; POM $O_3$= 0.85* Reference$O_3$ +0.56, $R^2$ = 0.93) (Figure S1). The measured vertical $O_3$ concentrations reported in this study were normalized to standard temperature and pressure conditions (25 °C, 1013.2 hPa).
- A Global Position System (GPS).
- Temperature, relative humidity, pressure, wind direction and wind speed sensors.

Moreover, another sounding was carried out on the 16/07/2015 at 11:00 UTC. A free balloon was
used with an instrumentation pack equipped with a Hy-CPC, a GPS, a temperature sensor, and a
relative humidity sensor. The pack was placed in an insulated box (Figure S4).

**Meteorological parameters**





30-minute meteorological data from 11 sites located throughout the study area in the proximity of
air quality monitoring sites were provided by Meteocat (Meteorological Office of Catalonia)
(Figure 1 and Table S3). Hourly average wind components were calculated and used in polar plots
with hourly $O_3$ and $O_x$ concentrations, by means of the OpenAir software (Carslaw, 2012). These
are bivariate polar plots concentrations are shown to vary by wind speed and wind direction as a
continuous surface.

**Modelling system for $O_3$**
The ambient $O_3$ concentrations were modelled using the ARAMIS (A Regional Air Quality Modelling
Integrated System) high resolution modelling system that integrates the Weather Research and
Forecasting model (WRF-ARW) version 3.1.1 (Skamarock et al., 2008) as a meteorological model,
the High Resolution Emission Model (HIREM) (Soler et al., 2004, 2011 and 2015), and the Models-3
Community Multiscale Air Quality Modelling System (Models-3/CMAQ) (Byun and Ching, 1999) as
a photochemical model. The modelling system was configured using four nested domains, D1, D2,
D3, and D4 with horizontal grids of 27, 9, 3 and 1 km, respectively. Initial and boundary conditions
for the meteorological model were taken for the European Centre for Medium-Range Weather
Forecast global model (ECMWF) with a 0.5º x 0.5º resolution, and the boundary conditions are
forced every 6 h, whilst for the photochemical model initial and boundary conditions model came
from a vertical profile supplied by CMAQ itself. Domains are run in one-way nesting and 24 h spin-
up was performed to minimize the effects of initial conditions for the inner domains. The output
data is saved every hour. ARAMIS is continuously updated, it has been extensively evaluated (Soler
et al., 2015) to simulate air quality over regional and local scales. In the present study the domain
D3 was used, which covers the area of interest.

**RESULTS AND DISCUSSION**
**Meteorological background and diurnal cycles of pollutants**
Two types of episodes that will be discussed in the following sections were identified concerning
the meteorological patterns and the $O_3$ concentrations recorded.
• Type A episode: Under "usual summer conditions", with the Azores High located west of
Iberia, and a ridge of high pressures extending into southern France, air masses in the
Western Mediterranean basin rotate clockwise (anticyclonic) during the day, changing the
direction at nighttime (Gangoiti et al., 2001), while the Atlantic gap winds (through the
Ebro and Carcassonne valleys), weaken during daytime due to inland sea breezes and
become strengthened during nighttime (Millán et al. 1997; Gangoiti et al., 2001, Gangoiti
et al., 2006 and Millán 2014). In such conditions, the air layers over the sea in front of
Barcelona tend to move within the south-westerlies during the day, following the
clockwise rotation, i.e. towards Southern France and the Gulf of Genoa, and within the
northerlies (towards Valencia) during the night. Thus, direct transport of $O_3$ and precursors
from the Fos-Berre-Marseille-Piombino (Livorno) area towards the BMA is weak or null.
However, indirect transport is more likely, first into the sea during nighttime conditions,
and then following the daytime south-westerlies for the combined coastal sea-breeze and
anticyclonic gyre at the coastal strip of Catalonia, which could bring a fraction of the
referred $O_3$ and precursors originated in southern France, together with those emitted at
the eastern coast of Iberia



- Type B episode: When the anticyclone establishes over Central Europe with relative low pressures to the West over the Atlantic, the flow pattern over the Western Mediterranean changeS: southerly winds blow at height over eastern Iberia, while at ground level, gap winds may weaken or stop the Mistral and the Tramontana winds in the Gulf of Lion, and Barcelona could then be directly affected by $O_3$ and precursors, coming with the easterlies blowing at the marine boundary layer (emissions from Corsica, Sardinia and Italy). However, under these atmospheric conditions $O_3$ levels did not reach the observed values found during episodes type A, and the $O_3$ daily records did not show the classical pattern of accumulation from one day to the next, characteristic of the highest $O_3$ episodes in the Western Mediterranean (Millán et al., 1997, 2000 and Castell et al., 2008a)

Under the above "usual summer conditions", Millán et al. (1991, 1996a, b, c, 1997, 2000, 2002), Gangoiti et al., (2001), and Castell, (2008a) demonstrated the vertical recirculation of $O_3$-rich masses in the western Mediterranean, with $O_3$ being formed from precursors transported inland by the combined up-slope and sea breeze winds. $O_3$ loaded air masses, elevated by topography and sea-mountain breezes will be transported back to the coastal area at a certain altitude during the day and accumulates in elevated stably stratified layers at the coastal areas during the late evening and night. During nighttime and at ground level $O_3$ depletion dominates mainly in urban and industrial centers, driven by reaction with new emissions, which at the coastal area are transported offshore within the stable surface drainage flows.

The synoptic atmospheric situation in July 2015 was characterized by an intense high pressure system over central and southern Europe during almost the whole month (Figure S5). Type A and B scenarios alternated, transporting warm air masses from North Africa towards higher latitudes by the anticyclonic dynamic and reaching extremely high temperatures in Europe. The stagnation of air masses induced a regional meteorological scenario in the area under study, characterized by local/regional re-circulations and sea-land breezes, both channelled by the complex topography. The flow pattern, together with the observed stably stratified layers developed up to a height of 2000-2500 m a.s.l. (Figure S5) associated with subsidence, enhanced the accumulation of pollutants and caused several pollution episodes in the north-eastern Iberian Peninsula. Coastal and pre-coastal locations (Barcelona) were mainly affected by daily sea breezes, starting blowing from the east (around 08:00 UTC) and turning progressively to south and southwest. The sea breezes were channelled through the valleys, which are mainly located following a north-to-south axis, and arrived to the monitoring stations predominantly from a southerly direction. However, during the night atmospheric conditions were much more stable with flow patterns dominated by land breezes from N-NW.

The VIC site was characterised by stagnant conditions during the day, reaching the maximum wind speed (4 m/s on average) at around 15:00 UTC, when sea-breeze intensity was at the highest (Figure 2). During the night very light winds blew from the north (Figure 2). During the periods 14-20/07/2015 (episode type A) and 03-06/07/2016 (episode type B) the sea breeze blew from 10:00 to 18:00 and 10:00 to 21:00 UTC, respectively; but in the first one the wind speed was higher (maximum of 2.7 m/s as an average for the period) and maximal at 14:00 UTC, whereas in the second wind speed was lower, with a maximum mean value for the period of 2.4 m/s at 17:00 UTC, but only 1.5 m/s at midday.

Averaged ground $O_3$ concentrations during type A episode recorded at VIC were clearly influenced by these wind patterns, showing a typical midday peak, followed by a higher peak at 13:00-14:00 UTC probably caused by the transport of BMA air masses by the breeze (Figure 2). Mean $O_3$ levels during this A episode reached 195 μg/m³ at 13:00 UTC. During type B episode averaged $O_3$ levels





were also very high (142 µg/m$^3$ at 14:00 UTC) but clearly lower than during the A episode (Figure
2).

Intensive surface measurements were only available for 10-17/07/2015 (when the mobile
laboratory was working at VIC). Average SO$_2$ levels for this period (included in the type A episode)
showed a similar daily pattern to that of O$_3$ (Figure 2) pointing to a probable Hewson's-type I
fumigation process (Hewson, 1964, Geiger et al., 1992) when midday convective flows that abate
to the surface a SO$_2$ rich layer accumulated in the limit of the boundary layer. Ground level
concentrations of BC, NO$_2$ and PM$_x$ showed a similar daily pattern driven by stagnation and traffic
rush hour, with maxima concentrations around 06:00 UTC (08:00 h local time, Figure 2). Finally,
extremely high concentrations of NH$_3$ (this is one of the most intensive farming regions of Spain
and mean values of the Vic Plain dosimeters reached 30 µg/m$^3$ NH$_3$ for 01-31/07/2015) followed
the typical midday maximum due to evaporative emissions from fertilisers, but the rapid increase
of the wind speed and dilution by the growth of the PBL thickness (see vertical profiles of
temperature, aerosols and O$_3$ at VIC in following sections) probably account for a reduction of
ground level NH$_3$ concentrations during the central hours of the day (Figure 2).
The varying diurnal and nocturnal air mass patterns in the Vic Plain are also shown by the PM$_{2.5}$
chemical composition. Figure S6 shows the 8-h concentration patterns of selected components
during the week period of 10-17/07/2015, including several days (14 to 17/07/2016) of the type A
episode defined above, affected by polluted air masses from the BMA.
In addition to the regionally transported O$_3$, concentrations of elemental carbon (EC) and traffic
and industry-related metals (including Zn, Cu, Pb, Sn and Sb) were notably enhanced at the end of
the week, and were attributed to local sources. This enhancement was most obvious during the
00:00-08:00 UTC period (Figure S6), under calm or northerly low wind (drainage slope winds)
carrying metallic pollutants from the Cu-smelter located 13 km to the north of Vic, and leading to
high Cu, Zn, Sn, W, Pb and Sb concentrations on the nights of 15 and 16/07/2015 (Figure S6). The
increase in EC was related to local traffic emissions during the morning rush hour as deduced from
the peaking MAAP BC concentrations during 05:00-08:00 UTC (07:00-10:00 h local time), up to 5
µg/m$^3$ hourly BC, when compared to 3 µg/m$^3$ recorded as maximum traffic rush hour
concentrations in the preceding days (data not shown). In contrast, the rise of organic carbon (OC)
concentrations observed during 08:00-16:00 UTC is attributed to the formation of secondary
organic aerosols (SOA).
Sulphate concentrations did not show any trend, as expected from secondary inorganic
components present in relatively homogeneous concentrations on a regional scale, whereas
nitrate (and in a minor proportion ammonium) concentrations increased during the evening
periods as a result of gas/particle partitioning (Figure S6). Interestingly, the stronger southerly
winds during the daytime in the second part of the week (see below) appear to have brought
polluted air from the BMA as signalled by slightly higher V concentrations (tracer of fuel oil
combustion); but also the fumigation from high strata (polluted for air masses that were injected
the previous day-s) might account for these SO$_2$ and V increases.
The concentrations of mineral matter and all its components (Al, Fe, Mg, Li, Ti, Rb, Sr, Ti, As) were
constant during the week, with relatively higher concentrations in the 08:00-16:00 UTC samples
(Figure S6), indicating a higher resuspension caused by stronger afternoon winds. The increment
on the 15/07/2015 (08:00-16:00 UTC) was attributed to resuspension of local dust, given that the
occurrence of African dust outbreaks was not observed during this period.



The free sounding measurements carried out at 11:00 UTC on 16/07/2015 revealed stratified air
masses up to 3000 m a.g.l. (Figure S7). The vertical profiles of potential temperature, water vapour
and aerosol concentration distributions can be used for the identification of atmospheric layers
presenting different properties: a lower layer up to about 1100 m a.g.l. characterised by a relative
high aerosol concentration, well mixed and with a relative high and uniform water vapour content.
A clear discontinuity between 1100 to 1500 m a.g.l limits the mentioned lower layer and a series
of stably stratified layers up to a height of 3000 m a.g.l. This layering of pollutants is probably
related to the development of regional and mesoscale convective cells driven by the combined
upslope and sea breeze flows developed the day before (Millán et al., 1997).

**$O_3$ and $O_X$ episodes**
Figure S8 shows the average $O_3$ and $O_X$ ground level concentrations recorded in July 2015 in the
study area at the XVPCA air quality monitoring network and with the passive dosimeters. $O_X$
maximum concentrations were recorded at the Vic Plain area and in the coastal sites at the
northeast of Barcelona. This is due to the high $O_3$ concentrations in Vic and to a higher proportion
of primary $NO_2$ (emitted mainly from diesel engines, and not formed in the atmosphere from NO
titration by $O_3$) in the coastal cities, respectively.
In July 2015, the $O_3$ hourly information threshold was exceeded a total of 74 times at the XVPCA
stations of Catalonia, 57 taking place in the Vic Plain stations (TON, VIC and MAN), and 69 in the
surrounding areas (pre-Pyrenees, High Llobregat river and Montseny).
Figure 3 shows hourly $O_3$ concentrations for the study period from selected monitoring sites. $O_3$
concentrations recorded at a coastal (Begur, BEG; blue, 200 m a.s.l.) and a remote inland western
pre-Pyrenean site (MSC, light green, 1570 m a.s.l.) (Figure 3a) show relatively narrow diurnal
variations and multiday episodes, with low or enhanced concentrations, according to
meteorological fluctuations. $O_3$ variations at the coastal BEG are opposed to those at the inland
MSC: As shown by the polar plots from Figure 4, relatively low $O_3$ concentrations (but still high in
absolute terms) were recorded at the BEG coastal site (easternmost site in this figure) when the
wind blows from the sea, whereas polluted air masses are transported towards the inland remote
MSC (westernmost location in the figure) site under the same meteorological conditions.
Conversely, when westerly winds blow, the inland remote MSC site received relatively clean air
masses with low $O_3$ (Figure 4), which are progressively loaded with regional pollution as these are
transported towards the coastal BEG site.
Data from two urban background sites of Barcelona (PLR and CTL, 81 and 5 m a.s.l., grey and black
in Figure 3b) show evidence of a high nocturnal $O_3$ consumption, with differences due to local $NO_x$
traffic emissions. Following the transport of air masses by combined breezes, the two sites located
in the northern periphery of the BMA, along the Besòs river valley (GRA and MON, 140 and 33 m
a.s.l., orange and yellow in Figure 3c; 20 and 6 km from BMA in NE an NNE directions, respectively)
show local $O_3$ production, with higher midday concentrations, while very low nocturnal levels
reflect again the intensive $O_3$ consumption (in a densely populated basin). $O_3$ concentrations were
closer between GRA and MON than between the two Barcelona urban sites (PLR and CTL).
Relevant $O_3$ net production and fumigation can be readily seen in the inner Vic Plain (TON, VIC and
MAN; 620, 498, 460 m a.s.l.; red, pink and violet in Figure 3d; 45, 55 and 62 km from BMA in a NNE
direction, respectively) as well as at the remote eastern pre-Pyrenean site of Pardines (PAR,
brown, 1226 m a.s.l, 102 km from BMA in a NNE direction), where $O_3$ formation and fumigation
seems to have already reached its maximum, and similar $O_3$ concentrations were recorded at all



sites during the midday increase. This suggests that the intensity of $O_3$ formation and fumigation
was clearly reduced in the Vic Plain-Pyrenees transect with respect to the Barcelona-Vic Plain (an
intermediate production place would be MSY (720 m a.s.l.; green in Figure 5, 39 km from BMA in a
NE direction). Polar plots of GRA, TON, MSY, VIC, MAN show clearly that the highest $O_3$ levels were
recorded with wind blowing from the Direction where BMA is located (Figure 4).
As it can be observed in Figures 3 and 5, during two periods (01-02 and 07-20/07/2015) $O_3$
concentrations increased progressively from the Barcelona city towards the northern BMA (GRA
and MON), the intermediate MSY regional background area and towards the northern Vic Plain
sites; and from there it slightly decreased towards the eastern pre-Pyrenees (PAR) following the
midday-afternoon combined breeze transport (Figures 3 and 5). During these days, no exceedance
of the information threshold was produced in the urban environment; only sporadic
measurements above the human protection target value were recorded in the close surroundings
of Barcelona. However, frequent exceedances of both thresholds were recorded in a regional
transport context towards the north of the BMA.
While differences in $O_3$ concentrations between TON, GRA, MSY, BEG and CTL were observed
during the period 03-06/07/2015 (B type episode), $O_X$ concentrations show a very similar behavior
along the Vic Plain, both qualitative and quantitative (Figure 5, $O_X$ is not reported at BEG due to
the lack of $NO_2$ measurements). Conversely, in the period 07-20/07/2015 (that includes the A type
episode), characterized by a change in the synoptic conditions, differences in daily maximum $O_X$
values resemble the same behavior of $O_3$ alone, with a positive and marked inland gradient. $O_3$
concentrations at BEG, a coastal site far in the northeast were higher during the former period and
showed low intra-day variation, indicating a probable long range transport of polluted air masses
(Figure 5).
$O_3$ and $O_X$ concentrations at the regional background site (MSY, 720 m a.s.l., green in Figure 5)
depict also the meteorologically influenced patterns but with a clear overlapped and pronounced
daily fluctuation, with marked higher concentrations indicating $O_3$ generation from a regional
origin, especially on 01-02 and 07-20/07/2015 (Figure 5).
Diurnal $O_3$ concentrations in the Vic Plain (around 460-620 m a.s.l.) were markedly higher than at
the coastal (CTL, PLR) sites, and slightly higher than at mountain sites (MSY, PAR and MSC, from
720, 1226 and 1570 m a.s.l.) during the 01-02 and 07-20/07/2015 periods. The $O_3$ hourly
information threshold of 180 μg/m³ was exceeded 55 times in the Vic Plain (3 sites), with 50 of
these exceedances taking place during 01/07/2015 and 14-20/07/2015. For these exceedances, an
hourly contribution of up to 150 μg/m³ of $O_X$ (mostly $O_3$) both from fumigation of recirculated
return layers (injected at an altitude of 1500-3000 m a.g.l. in the prior day(s)), and from transport
and photochemical generation of $O_3$ of the BMA plume, might be estimated based on the
differences of the $O_X$ early afternoon maxima recorded at the coastal BMA sites and the ones in
the Vic Plain.

Type A episode (14-20/07/2015)
During this episode, a progressive time shift of the daily hourly $O_3$ and $O_X$ maxima was observed
from the Barcelona area (10:00 UTC, at CTL into the BMA) towards the metropolitan periphery
(11:00, at GRA), the intermediate mountain sites (13:00, MSY, 39 km from BMA), the Vic Plain
(12:00, 13:00 and 14:00, TON, VIC and MAN, 45, 55 and 62 km from the BMA, respectively) and
the northern pre-Pyrenean site (16:00, PAR, 102 km from BMA) (Figure 6). This variation points to
the process of $O_3$ and $O_X$ formation with a mean $O_X$ difference between the urban-coastal sites and





the Vic Plain hourly maxima of around 115 µg/m$^3$ for the TON site, with a maximum average $O_3$
hourly levels of around 200 µg/m$^3$. These $O_x$ differences are mostly due to $O_3$ differences (Figure
6). Accordingly, during these intense $O_3$ pollution episodes, more than 50% of the $O_3$ hourly
maxima concentrations are attributable to (i) $O_3$ contributions from the previously referred
surface fumigation of recirculated strata (over the VIC-MAN-TON area) containing the polluted air
masses injected the day before by complex topographic induced circulations; and to (ii) the local
$O_3$ generation and surface transport of the BMA plume into inland valleys. Attributing these $O_3$
exceedances to local/regional causes is also supported by the spatial distribution of the hourly $O_3$
maxima, the number of hourly exceedances of the information threshold, the time shift of the
exceedances at the different sites (as moving towards the north) (Figure 7), and the polar plots of
hourly $O_3$ concentrations pointing towards the BMA as main source region (Figure 4).
Thus during the A episode, $O_3$ has mostly a major local/regional origin (with $O_x$ maximum hourly
levels progressively increasing from 166 to 246 µg/m$^3$ from the BMA to the Vic Plain). The
concatenation of daily cycles of regional/long range recirculation of air masses and regional/local
$O_3$ production in the A episode accounted for the accumulation of $O_x$ and the consequent
exceedance of the hourly information threshold. Castell et al. (2008a) have already reported a
correlation between their 'recirculation factor' (a meteorological parameter devised to increase
with the concatenation of days with regional vertical recirculation of air masses) with the
occurrence of $O_3$ episodes in 2003. The relevance of these recirculations in originating these high
$O_3$ episodes in Southern Europe was highlighted already, not only by scientific papers by the
CEAM's team but also assumed by the European Commission (EC, 2004).
Figures 8 and 9 show results the vertical profiles (0-1100 m a.g.l.) of $O_3$ concentrations, particle
number concentrations for particles > 3 nm (red), 0.3-0.5 µm (blue), 0.5-1.0 µm (brown), ambient
temperature, relative humidity and wind direction, obtained at the beginning of the A type
episode (from 14 to 17/07/2015).
In the profiles from 07:06 to 08:21 UTC on the 14/07/2015, a boundary layer (150 to 250 m thick)
with relatively high levels of $N_3$ (0.8 to 2.0 10$^4$ #/cm$^3$) was differentiated from the free troposphere
(0.2 to 0.8 10$^4$ #/cm$^3$) (Figure 8). However, in the profile obtained from 09:42 to 10:52 UTC on
17/07/2015, the growth by convective turbulence accounts for a homogeneous boundary layer
and profile of $N_{0.3-0.5}$ below 1000 m a.g.l. (Figure 8). Inside the boundary layer nucleation occurred
(yellow to red areas in Figure 10 for 16/07/2015) regionally driven by photochemical processes.
Minguillón et al. (2015) showed the occurrence of these nucleation events into the mixing layer as
convective transport elevates and dilutes air masses from polluted areas under high insolation in
Barcelona. During 14-16/07/2015 nucleation episodes occurred occasionally, but only inside the
boundary layer. On the 17/07/2015 at 9:42-10:52 UTC new particle formation occurred probably
at relatively high altitudes, also inside the boundary layer, as deduced from the high $N_3$ levels
measured from 400 to 1000 m a.g.l., with concentrations reaching 1 10$^4$ #/cm$^3$, while
simultaneously low concentrations (< 0.3 10$^4$ #/cm$^3$) were measured at ground level (Figure 8).
This vertical gradient is not observed for the coarser particles ($N_{0.3-0.5}$ and $N_{0.5-1}$) and $O_3$ (Figures 8
and 9, for which relatively constant levels were measured inside the boundary layer), suggesting
new particle formation.
On 14/07/2015 07:06-08:21 UTC a well stratified atmosphere (Figure9) with both thermal and $O_3$
layers is observed, with a general upward increasing trend for $O_3$ from 40 µg/m$^3$ at ground level to
150 µg/m$^3$ at 1000 m a.g.l. reflecting the effect of surface depletion by NO titration and by
deposition during the night (Figure 9). From 13:49 to 15:03 UTC on the 14/07/2015 (Figure 9) the
vertical profile changed substantially, with an already unstable atmosphere near the ground,





showing very high surface $O_3$ concentrations of 217 µg/m$^3$ that increase up to 330 µg/m$^3$ in a layer
around 100 m a.g.l., decreasing again through an upper layer with values of 240 µg/m$^3$ until 300 m
a.g.l. (where measurements were not available due to instrumental problems). This 100-200 m
a.g.l. very high $O_3$ layer agrees with the modelled $O_3$ concentrations for the study area (Toll and
Baldasano, 2000; Barros et al., 2003; Gonçalves et al., 2009) and reflects elevated $O_3$
concentrations due to local production and transport of $O_3$, that decrease from 100 m a.g.l. to the
surface due to its titration, consumption and deposition. On 15 and 16/07/2015, a similar upward
increasing $O_3$ gradient was observed in the early morning (Figure 10). On 17/07/2015 7:39-08:40
UTC $O_3$ concentrations were relatively constant, but showing also a strongly stratified profile, in
the range of 100-165 µg/m$^3$ in the lower 500 m. In the last profile from 09:42 to 10:52 UTC, $O_3$
concentrations increased from 140 to 200 µg/m$^3$ from 200 to 1000 m a.g.l., but again a maximum
close to 200 µg/m$^3$ was observed at the same height around 100 m a.g.l. (Figure 9).
Thus, vertical profiles of the type A episode are characterised in the early morning by a strong
stratification, showing low ground level $O_3$ concentrations, due to low production (low insolation)
and/or consumption (titration and deposition), and increasing concentrations with altitude. This
variation is related to prevailing meteorological conditions enhancing local recirculation or larger
scale transport with high $O_3$ masses injected (the day before) at certain altitudes by vertical
recirculations into the residual layer, above the nocturnal surface stably stratified boundary layer.
Nevertheless, during specific days, homogeneous $O_3$ vertical profiles up to 1000 m a.g.l. (the
maximum height reached with captive sounding) were also evidenced, but probably not
maintained at higher levels (where we were not able to measure with our system). Thus, as shown
by the 4500 m profile measured with the free sounding on 16/07/2016 (Figure S7), high PM (and
probably $O_3$) strata are present between 1500 and 3000 m a.g.l., these being probably the polluted
air masses injected the day before in the northern mountain ranges and recirculated to the coast at
certain altitudes (see modelling outputs below). On the other hand, with constant southerly winds
(from the coastal area to the Vic Plain) usually associated with the combined sea-breeze and up-
slope flows, $O_3$ was enriched in the lower 100-200 m atmospheric layer, generated by the
intensive local photochemical production. $O_3$ concentrations reached maximal values (up to 330
µg/m$^3$) on the top of this layer, while they decreased at lower heights by titration and deposition,
although hourly levels of 225 µg/m$^3$ were still recorded. These results are consistent with the
gradient of $O_3$ concentrations between the Vic Plain (around 500 m a.s.l.) and the MSY mountain
site (720 m a.s.l., and more close to the sea) during the episodes, (Figures 5 to 7). At higher
altitude, $O_3$ concentrations decreased but were still high (150-240 µg/m$^3$) due to the $O_3$ formation
in air masses constantly transported from the coastal area, which also incorporates $O_3$ and
precursors recirculated the day before, as it is shown next.
Interesting results are also obtained by comparing the vertical profiles of BC and $O_3$ (Figure 11). BC
is a tracer of local primary pollution at ground level, and of the potential transport and
stratification of regional/local primary pollutants (together or not with regional $O_3$) when present
at high altitude. On 14/07/2015 07:06-08:21 UTC, at 350 m a.g.l. (and similarly on 15-17/07/2015
but at varying heights, 100-350 m a.g.l.) a clear discontinuity is evidenced with sudden and
simultaneous decreases of BC and $O_3$ above these heights. The relatively high BC levels within the
lower layer suggest the nocturnal accumulation, while $O_3$ appears in strata (with low values near
the ground due to titration and deposition) and with a high concentration just above that level
(350 m), now with low BC concentrations. There is a further upward decrease of BC and an
increase of $O_3$ up to the limit o the sounding (870 m).
The occurrence of an $O_3$ maximum layer around 100-200 m a.g.l., on top of the nocturnal stably
stratified boundary layer reinforces the idea of an important local production contributing to an





upward increase of $O_3$ inside the layer. Finally, at the highest altitudes reached in this study (900-
1000 m a.g.l.), BC and $O_3$ concentrations were often anti-correlated or unrelated, possibly more
related with aged air masses re-circulated within the whole region and with a mixed origin:
including local-to-regional sources and more distant over the W-Mediterranean.
Figure 12 shows mean $O_3$ hourly concentrations recorded at VIC for the episodes A and B, as well
as mean wind speed and directions. Mean hourly concentrations are characterized by an increase
until 10:00-11.00 UTC, followed by an inflexion point and a more marked increase, with a maxima
between 13:00 and 14:00 h UTC, and then a progressive decrease, more marked in the episode. As
stated above, processes contributing to increase levels were attributable (ordered by importance)
to fumigation, photochemical production and transport of high $O_3$ air masses, all controlled by
insolation. Millán et al. (2000) described this characteristic diurnal $O_3$ pattern typically for inland
valley stations (as in our case around 75 km from the coast), where the first $O_3$ increase is
attributed to $O_3$ contributions from surface fumigation of high recirculated return strata as well as
from the arrival of higher $O_3$ air masses transported by sea-breeze and the local photochemical
production from precursors. On the other hand, the second $O_3$ concentration 'hump' is coincident
with maximum wind speed and probably corresponds to a more intensified sea breeze transport
compared with local photochemical formation and fumigation. Figure 12 shows that the two $O_3$
increases (and consequently the contributions from the 3 above processes) are more pronounced
in the type A compared to the type B episode, and that the second maxima (more associated to
inland surface transport by sea breeze) is wider, coinciding with a shift of the maxima wind speed
towards the late afternoon, in the B episode.
Modelling outputs for the A episode points to light winds from the south, transporting pollutants
from the BMA towards northern areas (including the Vic Plain), and triggering the hourly $O_3$
exceedances under the effect of the sea and land breeze transport. Thus, Figures 13 and 14 show
the horizontal wind vector at 10 m a.g.l. and $NO_2$ and $O_3$ concentrations both at ground level and
at a height above the surface layer, at different hours for two representative days of the type A
and B episodes, respectively. During the type A episode day (15/07/2015), the effect of the land
breeze transport accumulates $NO_2$ over the sea during the night, starting intense $O_3$ production
when sun rises and sea breezes start the inland transport. Maximum concentrations of $O_3$,
exceeding 180 $\mu g/m^3$ were calculated by the model and measured at the stations located in the
Vic Plain (TON, VIC and MAN, Figure 14), although the model overestimated maximum $O_3$
concentrations in TON and VIC and delayed the hourly maximum value in all stations. The vertical
distribution shows an important accumulation of around 110 $\mu g/m^3$ trapped in a reservoir layer at
around 1500 m a.s.l. during the night (Figure 15), which will fumigate downwards into the new
developing mixing layer during the following hours. Local $O_3$ production from fresh precursors
accumulated during the night in the stably stratified surface layer and then progressed inland
along the midday hours. This results in an $O_3$ enriched plume within a layer of 1000-1500 m depth
in the late afternoon, following the model results (Figure 15). This mixing layer also incorporates
$O_3$ from upper reservoir layers after fumigation during the inland travel. The $O_3$ located at upper
levels can re-circulate back into the sea and it will be potentially available to be transported inland
(Millán et al., 1997 and 2002), to re-start a new cycle the following day.

Type B episode (03-06/07/2015)
As opposed to episode A, during the type B episode and 22-31/07/2015, despite the high $O_3$ and
$O_X$ concentrations, these concentrations were very similar in the urban and remote coastal sites
and all along the northern sites, including the Vic Plain. Hence, the averaged $O_X$ hourly
concentrations of all the study sites were close to those at the coastal urban site in Barcelona CTL





(and in the case of the $O_3$ close to the remote coastal site of BEG) compared with the large
differences reported for the A episode (Figure 6). The high $O_X$ peak measured at the urban site
during the mornings of the B period (Figure 6) and from 8:00 to 10:00 UTC in the average hourly
patterns (Figure7) is due to the contribution of primary $NO_2$. According to Carslaw et al. (2016) the
Euro 1 to Euro 2 diesel engines in Europe (early 1990s) emitted 5-10% of primary $NO_2$ and 90-95%
of NO, whereas the Euro 4 to Euro 5 equivalent engines (2004 and 2009 onwards) emit 16-29% of
primary $NO_2$ and 71-84% of NO.
Also as opposed to the A episode, during the B episode, $O_X$ levels varied in a very narrow range
from East (coastal) to West (mountains, MSC site) and from South (BMA) to North (Vic Plain) and
at different heights (from Barcelona and BEG at sea level to MSC at 1570 m a.s.l.). Following the
results of the model in the Figures 13 and 15, $O_3$ does not re-circulate around the region in this
period. There is no accumulation from one day to the next in reservoir layers located over the
region. Southerly winds blow at height during the whole period and the combined sea breeze and
upslope winds developed at lower layers during daytime, after coupling with the southerlies, vent
out the $O_3$ production and the rest of pollutants to the north. The circulation is open, as opposed
to the type A episode, which show a closed circulation (it is never completely closed) (Millán et al.,
1997, 2000). Unfortunately, vertical profiles of $O_3$, UFP, PM and BC profiles were not obtained for
this episode.
Model outputs also evidence a net night and early morning transport of $O_3$ at lower layers from
east and north-east during the B episode, supporting the hypothesis of a regional transport from
Southern France, advecting aged air masses to the whole region, while $O_3$ and its precursors from
the BMA were transported during the morning to the south-west regions (Figure 13) giving rise to
hourly $O_3$ exceedances in some stations situated in this area. Figure 13 also shows that during this
episode (03/07/2015) the combined sea-breeze and upslope wind transported $O_3$ and precursors
to the western pre-Pyrenees area, and values lower than 180 $\mu g/m^3$ were measured and modelled
in all monitoring stations and mainly in the Vic Plain. The vertical distribution of $O_3$ also shows
relatively low concentrations over most of the domain (Figure 15).

**CONCLUSIONS AND IMPLICATIONS FOR AIR QUALITY**
Very high levels of $O_3$ were recorded in the plains and valleys of the northern regions surrounding
the Barcelona Metropolitan Area (BMA) during July 2015, where 69 out of the 74 exceedances of
the hourly $O_3$ information threshold measured in the entire air quality monitoring network of
Catalonia were recorded. This represents a major environmental problem for which air quality
managers must implement European and national legislation.
Both experimental measurements and modelling exercises suggest that these very intense $O_3$
episodes were originated by the concatenation of daily cycles of vertical recirculation of air masses
that accumulated photochemically generated pollutants (Millán et al., 1997, 2000, 2002; Gangoiti
et al., 2001 and Castell et al., 2008a, Valverde et al., 2016), favoured by the high BVOCs and
anthropogenic $NO_X$ emissions in the BMA region. The lower 1000 m a.g.l. were highly enriched in
$O_3$ by fumigation from precursors and $O_3$ located at upper levels (1500-3000 m a.g.l). Additionally,
local contributions of $O_3$ to these episodes were also demonstrated by soundings of the lower
layers (0-1000 m a.g.l.). Thus, slightly higher concentrations of $O_3$ were measured in stations
located at the plains and inland valleys than at higher altitudes (up to +30-40/$\mu g/m^3$ added to
180$\mu g/m^3$ reached in the mountain sites), due to the local photochemical production from fresh
precursors emitted during the night and early morning, and their channelling within the combined



upslope and sea breeze circulation that transports $O_3$ and precursors from the BMA. Thus vertical
profiles identified a high $O_3$ layer 100-200 m a.g.l., produced by these local processes and also by
the high deposition and titration of $O_3$ at the lower 100m depth layer. In our (mostly rural) study
low concentrations of ultrafine particles were recorded at this high $O_3$ 100-200 m a.g.l. layer and
nucleation episodes were only detected into the boundary layer, most of the days at the lower
atmospheric levels.
Two types of high $O_3$ episodes (A and B) were identified in the area:
Type A episode: Characterized by major local/regional $O_3$ recirculation, fumigation, production and
transport, superimposed on the typical regional/long range transport, and by the occurrence of
major exceedances of the $O_3$ information threshold (14-20/07/2015). Surface fumigation from
high $O_3$ return (to the sea) layers injected the day(s) before at altitudes of 1500-3000 ma.g.l., and
recirculated over the VIC-TON-MAN area, as well as direct surface transport and formation of
local/regional polluted air masses (with $O_3$ and precursors) from the BMA towards the north,
decisively contributed to these exceedances. Thus, this atmospheric scenario is governed by poor
ventilation under local breeze circulations and vertical recirculation of air masses over the study.
Type B episode: With a major regional transport $O_3$ contribution, yielding similar $O_X$ levels at all
monitoring sites of the study area, with the arrival of aged air masses from the east/northeast
(high $O_3$ levels entering through the coast), but without major transport from BMA to the Vic Plain
(3-6/07/2015), and without vertical recirculation of air masses over the study area. When sunlight
activates atmospheric photochemistry in the early morning, the northern regions were loaded
with air masses with lower content of $O_3$ precursors, since air masses were transported from the
BMA to the southwest (parallel to the coast) from 00:00 to 09:00 h UTC. The combined breeze at
midday favored the transport towards the northwest, rather than to the north, as described for
the type A episode. In addition the aged air masses are not vertically recirculated and leave the
region towards the north-east (to France). Thus, $O_3$ concentrations were still relatively high
(exceeding 120 μg/m³ but below 180 μg/m³) due to local production from fresh precursors and
background $O_3$ contributions from the western Mediterranean, but not enough to exceed the
information threshold.
From the perspective of possible precursor abatement strategies, direct mitigation measures at
the BMA would have had a minor effect on $O_3$ concentrations at the Vic Plain area during the type
B episode. However, during the type A episode, a reduction of $NO_x$ and/or VOCs emissions in the
BMA, some days before and during the episode, might have an effect on $O_3$ concentrations
recorded in the Vic Plain. Nonetheless, due to the non-linearity of the $O_3$ generation processes,
sensitivity analysis with high resolution modelling is necessary to evaluate the possible effects in
terms of absolute concentrations.
The use of $O_X$ data from strategically selected monitoring sites in the east coast, western and
central mountain areas, urban background sites of the BMA and sites in the Vic Plain, tracking the
natural routes of pollutant transport, is a useful tool to assess the different regimes leading to high
$O_3$ concentrations, and to differentiate between type A and type B episodes, with important
implications in the design of potential abatement strategies.
To implement potential (and difficult) abatement strategies two major key tasks are proposed:
1. Meteorological forecast from June to August to predict recirculation episodes in order to
apply abatement measures for $O_3$ precursors before a recirculation episode starts. As state



above, the relevance of these recirculations in originating these high $O_3$ episodes in
Southern Europe was assumed already by the European Commission in 2004 (EC, 2004).
2. Sensitivity analysis with high resolution modelling to evaluate the effectiveness of $NO_x$ and
VOCs abatement measures on $O_3$ reduction.

**ACKNOWLEDEGMENTS**

The present work was supported by the Spanish Ministry of Economy and Competitiveness and
FEDER funds under the project HOUSE (CGL2016-78594-R), by the Generalitat de Catalunya
(AGAUR 2015 SGR33 and the DGQA). Part of this research was supported by the Korea Ministry of
Environment through "The Eco-Innovation project". The participation of University of Marseille
and University of Birmingham was partially supported by two TNA actions projects carried out
under the ACTRIS2 project (grant agreement No. 654109) financed by the European Union's
Horizon 2020 research and innovation program.. The support of the CUD of Zaragoza (project CUD
2013-18) is also acknowledged. We are very thankful to the Generalitat de Catalunya for supplying
the air quality data from the XVPCA stations, to METEOCAT (the Meteorological Office of
Catalonia) for providing meteorological data and to the IES J. Callís and the Meteorological Station
from Vic for allowing the performance of the vertical profiles and mobile unit measurements,
respectively.

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





**FIGURE AND TABLE CAPTIONS**
Figure 1. Top: Study area and location of monitoring sites (regional air quality monitoring sites
XVPCA, dosimeters, meteorological stations and vertical measurements). BMA: Barcelona
Metropolitan Area. Bottom: Topographic profiles across the study area, red arrows point to the
valleys connecting BMA with the Bic Plain and the Pre-Pyrenean regions.
Figure 2. Mean hourly (UTC) values for meteorological parameters and gaseous and particulate
pollutants for 03-06, 14-20 and 10-17/7/2015 recorded at VIC with the laboratory van (10-
17/07/2015), with the exception of $O_3$, measured at the same location but by the permanent VIC
XVPCA station, and the meteorological data obtained from Gurb station (Meteocat) located 1 km
to the north of VIC.
Figure 3. $O_3$ hourly concentrations recorded at the coastal (BEG, blue) and remote inland western
pre-Pyrenean (MSC, clear green, 1570 m a.s.l.) sites, 2 urban background sites of Barcelona (PLR,
CTL, grey and black), 2 urban sites in the northern periphery of the Barcelona's metropolitan area
(GRA, MON, orange and yellow), the inner Vic Plain sites (TON, VIC and MAN, red, pink and violet)
and the remote eastern pre-Pyrenean site of PAR (brown), along July 2015.
Figure 4. Polar plots of hourly $O_3$ concentrations in the real-time measurement sites.
Figure 5. $O_3$ and $O_X$ ($O_3$+$NO_2$) hourly concentrations recorded at the coastal (BEG, blue, at this site
only $O_3$ is available because the lack of $NO_2$ measurements), an urban background site of
Barcelona (CTL, black), an urban site in the northern periphery of the Barcelona's metropolitan
area (GRA, orange), the intermediate inland rural site of MSY (720 m a.s.l., green), and the inner
Vic Plain site (TON, red) along July 2015. The pink and blue squares mark the A and B $O_3$ and $O_X$
episodes distinguished in this study, respectively.
Figure 6. Mean hourly levels of $O_3$ and $O_X$ ($O_3$+$NO_2$) for sites located in a south and coast to north-
inland (CTL, GRA and BEG, and MSY, TON, VOIC, MAN and PAR, respectively) following the inland
transport of pollutants and maxima time shift according to this breeze transport (right) for the
periods 3-6/07/2015 (B type episode, left) and 14-20/07/2015 (A type episode, right). Time is UTC.
Figure 7. Top: Hourly $O_3$ maxima (and number of hours exceeding 180 µg/m$^3$) in the study sites
with real time $O_3$ measurements (shadowed areas indicate 2 different degrees of exceedances, 1-3
h and 13-23h). Bottom: Frequency of occurrence of hourly (UTC) $O_3$ exceedances of 180 µg/m$^3$
along the day; both for July 2015.
Figure 8. Vertical profiles of particle number concentrations for particles >3 nm (red, $N_3$), 0.3-0.5
µm (blue, $PM_{0.3-0.5}$), 0.5-1.0 µm ($PM_{0.5-1}$) and wind direction obtained with the tethered balloon
measurements on 14 and 17/07/2015.
Figure 9. Vertical profiles of $O_3$, temperature and relative humidity obtained with the tethered
balloon measurements on 14 and 17/07/2015.



Figure 10.Time variation of altitude, temperature, relative humidity, $N_3$, particle number size
distributions and $O_3$ concentrations during the tethered balloon measurements on 16/07/2015. 1
to 3 illustrate the nucleation episode recorded at surface level with particle number size
distributions, and 4 the typical regional background size N size distribution at around 300 m over
the ground.
Figure 11. Vertical profiles of BC (5 min time resolution) and $O_3$ (10 seconds time resolution) at
VIC.
Figure 12. Mean hourly $O_3$ concentrations, and wind speed and wind direction for the episodes A
and B, showing higher levels in the A episodes for the two $O_3$ maxima.
Figure 13. Maps of simulated $NO_2$ and $O_3$ concentrations at ground level and 1000m a.g.l., and
horizontal wind fields at 10 m a.g.l. for selected hours on 03/07/2015.
Figure 14. Maps of simulated $NO_2$ and $O_3$ concentrations at ground level and 1000m a.g.l., and
horizontal wind fields at 10 m a.g.l. for selected hours on 15/07/2015.
Figure 15. Spatial distributions of simulated $O_3$ concentrations and wind field vectors in the south–
north vertical cross-section for different hours on 03 and 15/07/2015.



Figure 1



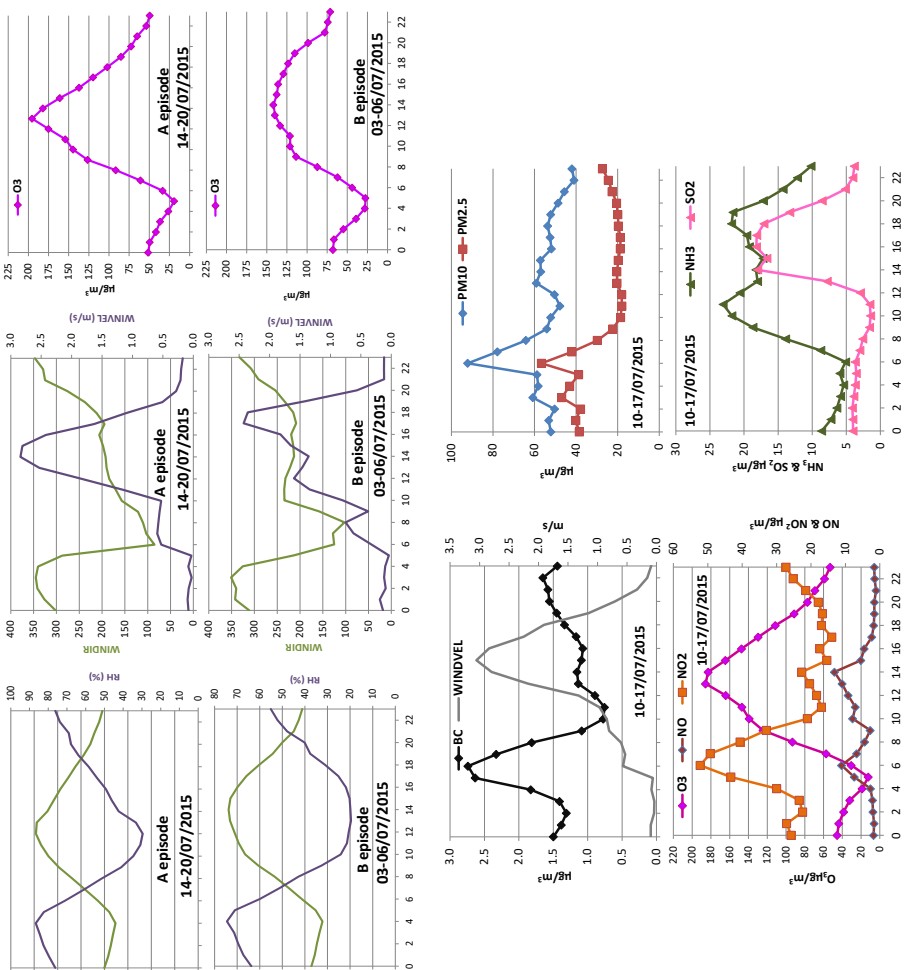

Figure 2





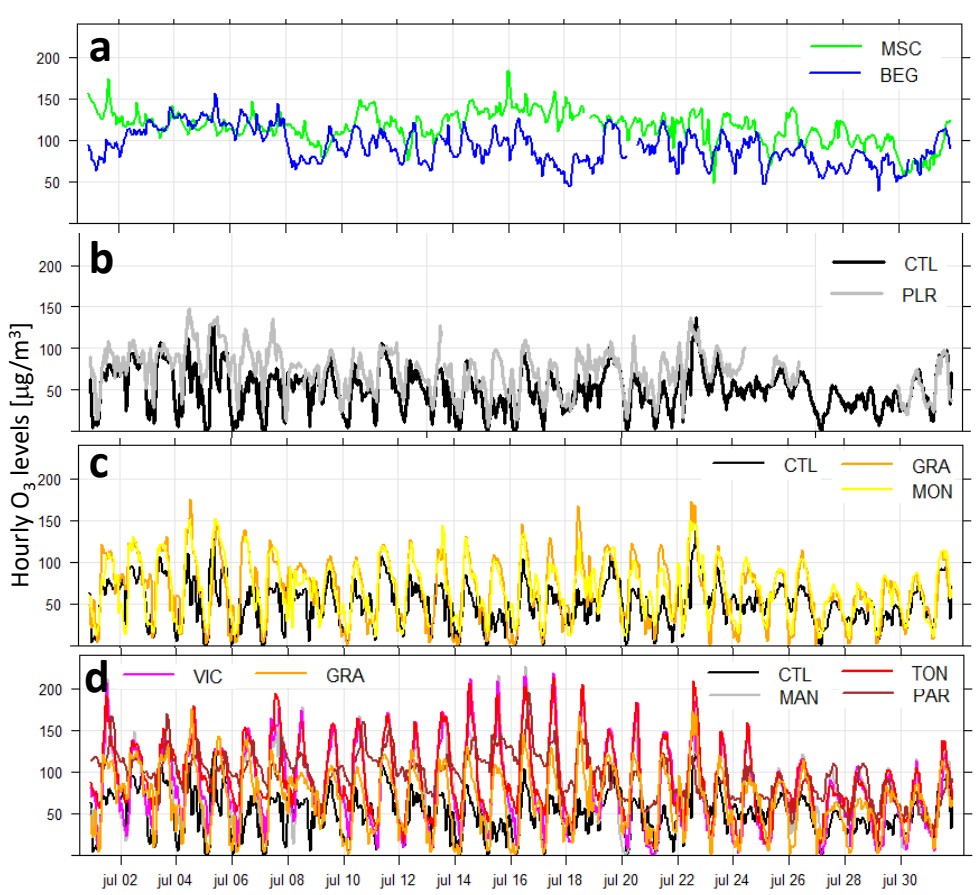

FIGURE 3





FIGURE 4





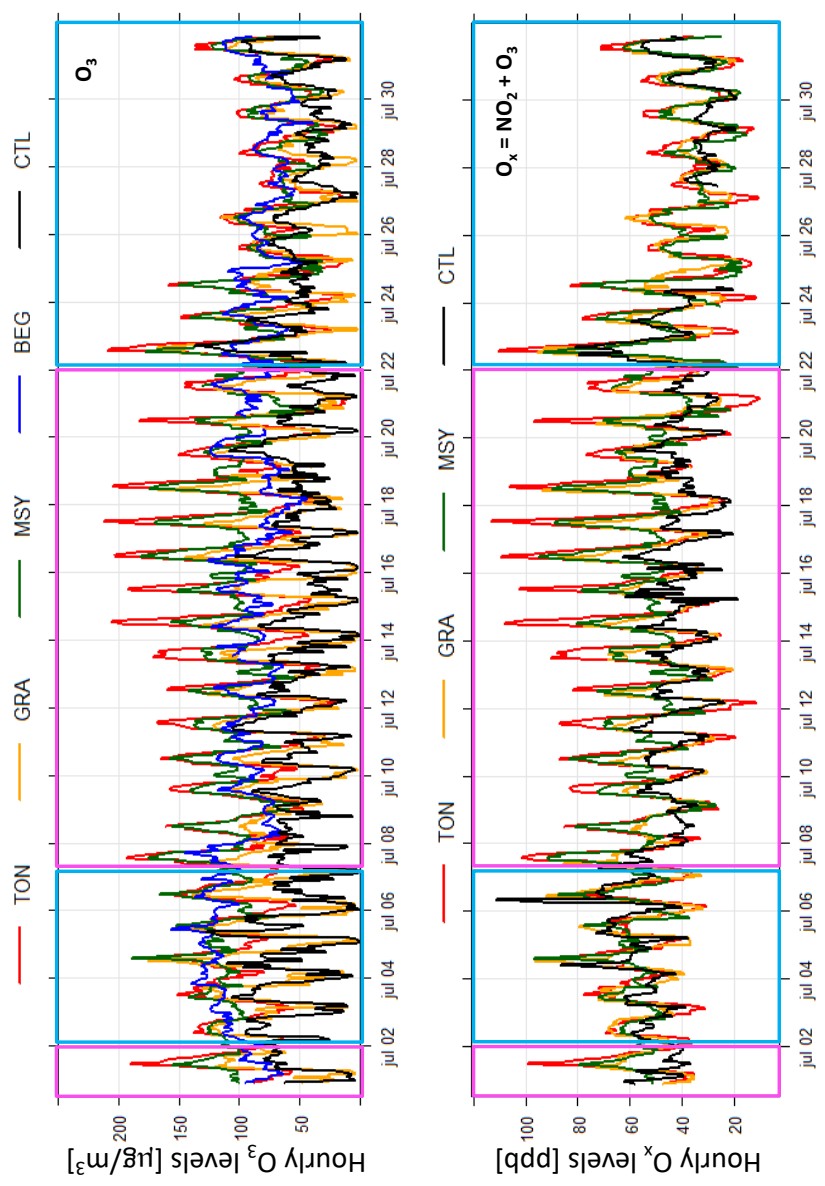

FIGURE 5





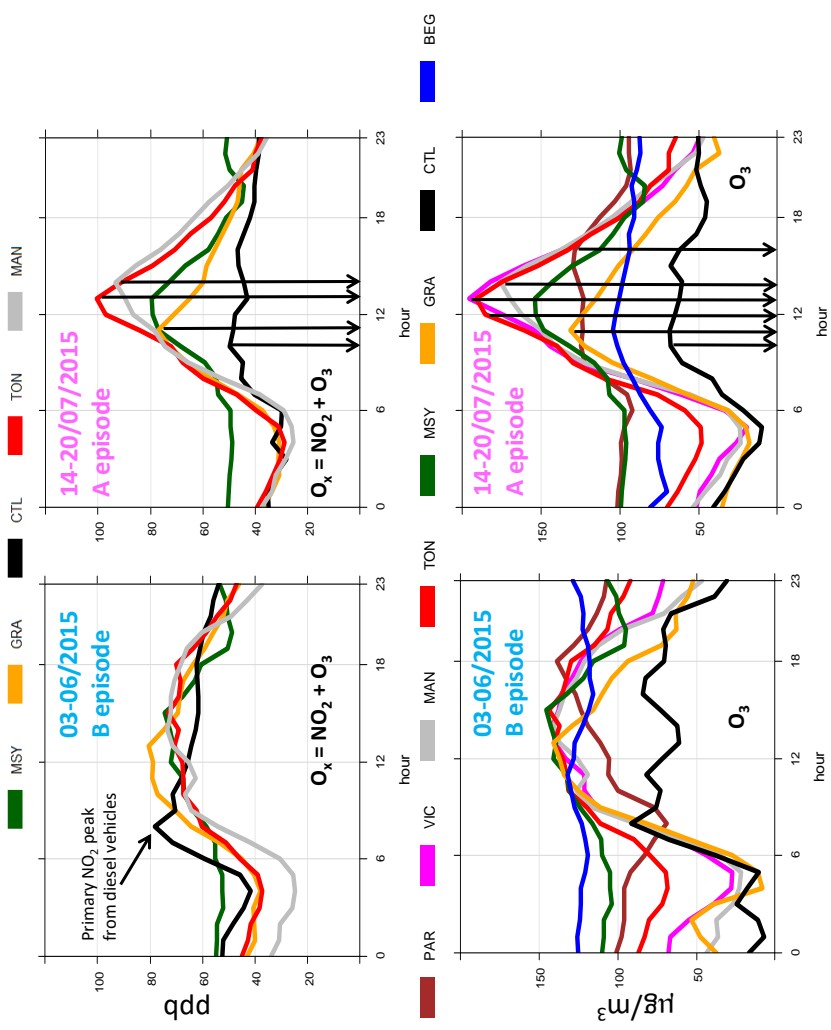

FIGURE 6.





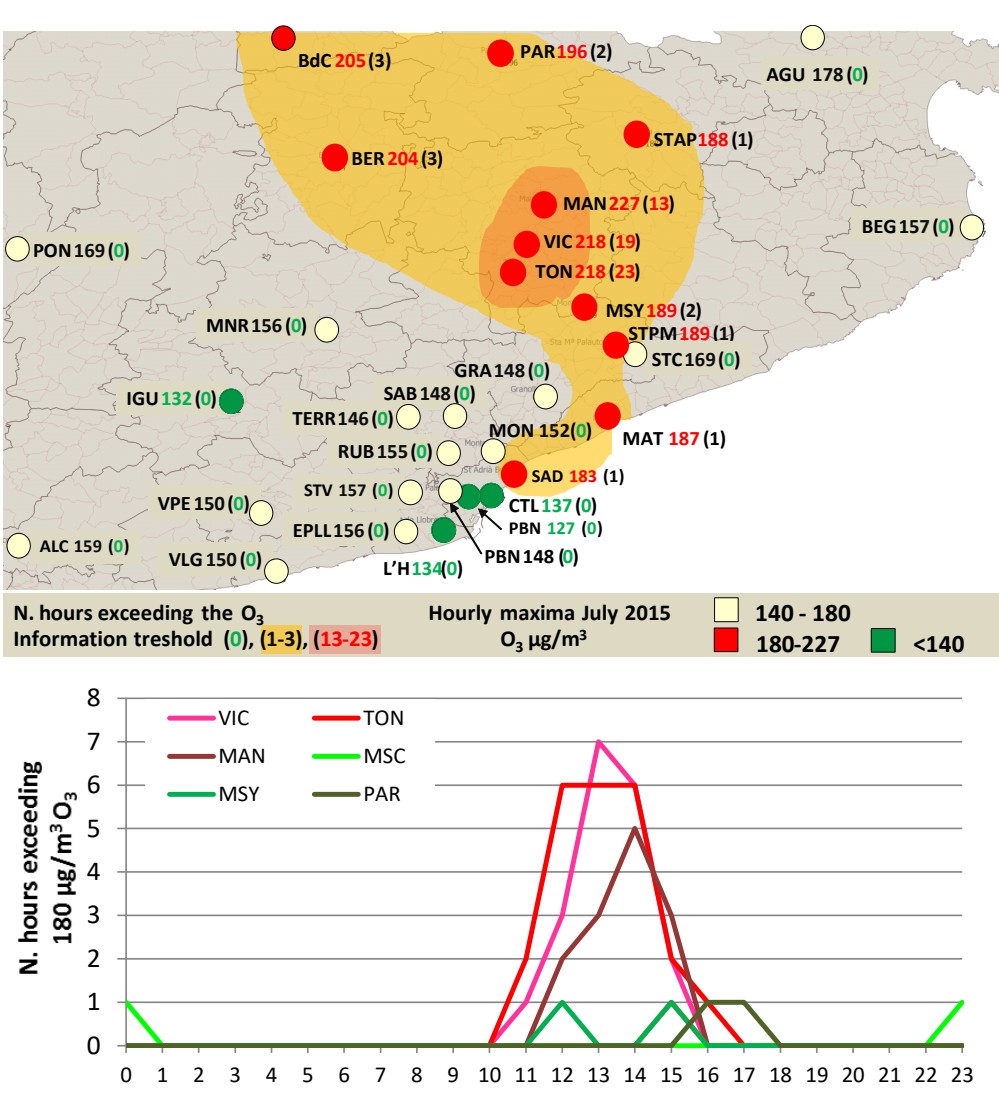

FIGURE 7.



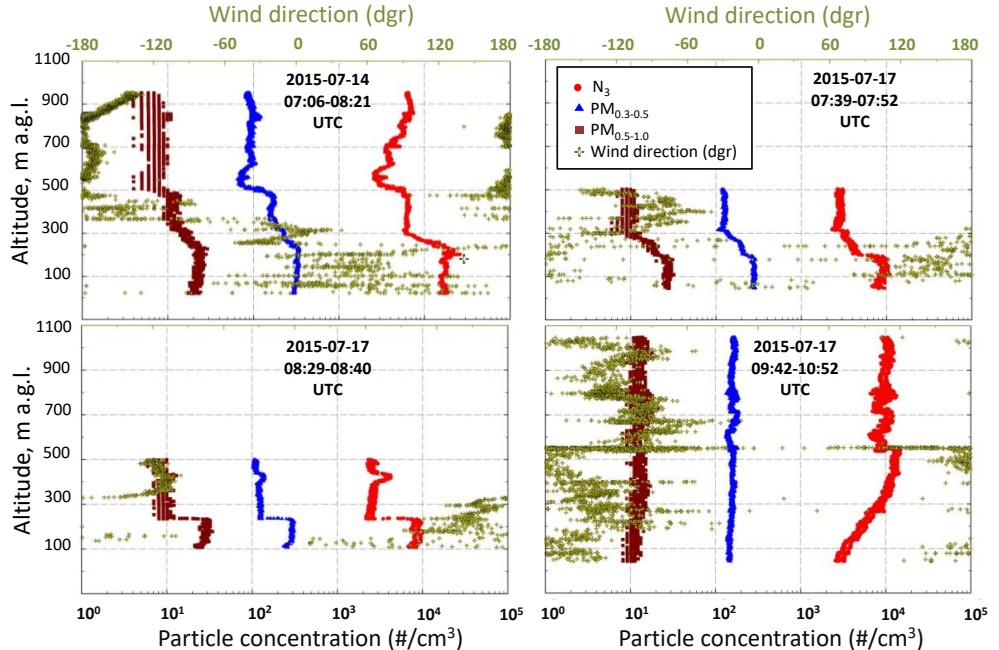

FIGURE 8.

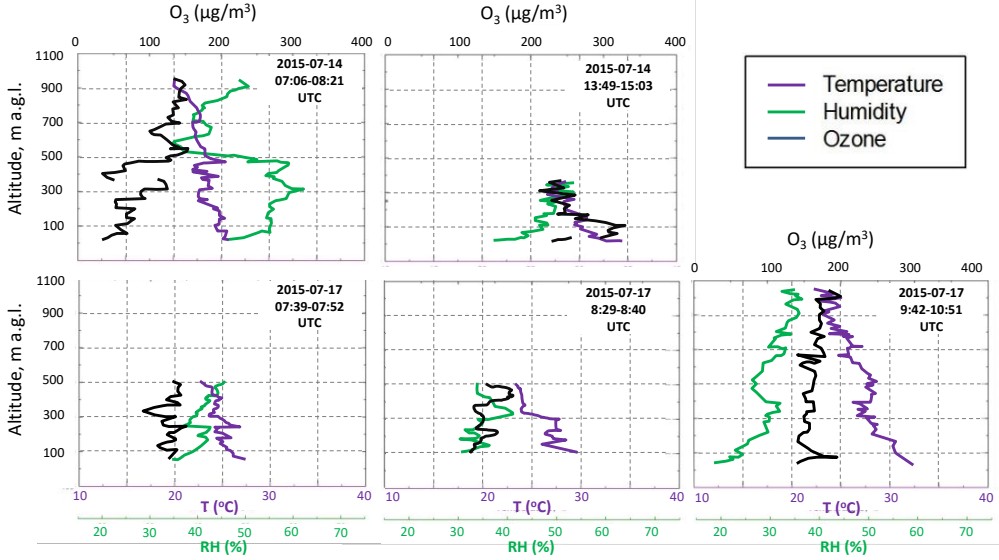

FIGURE 9



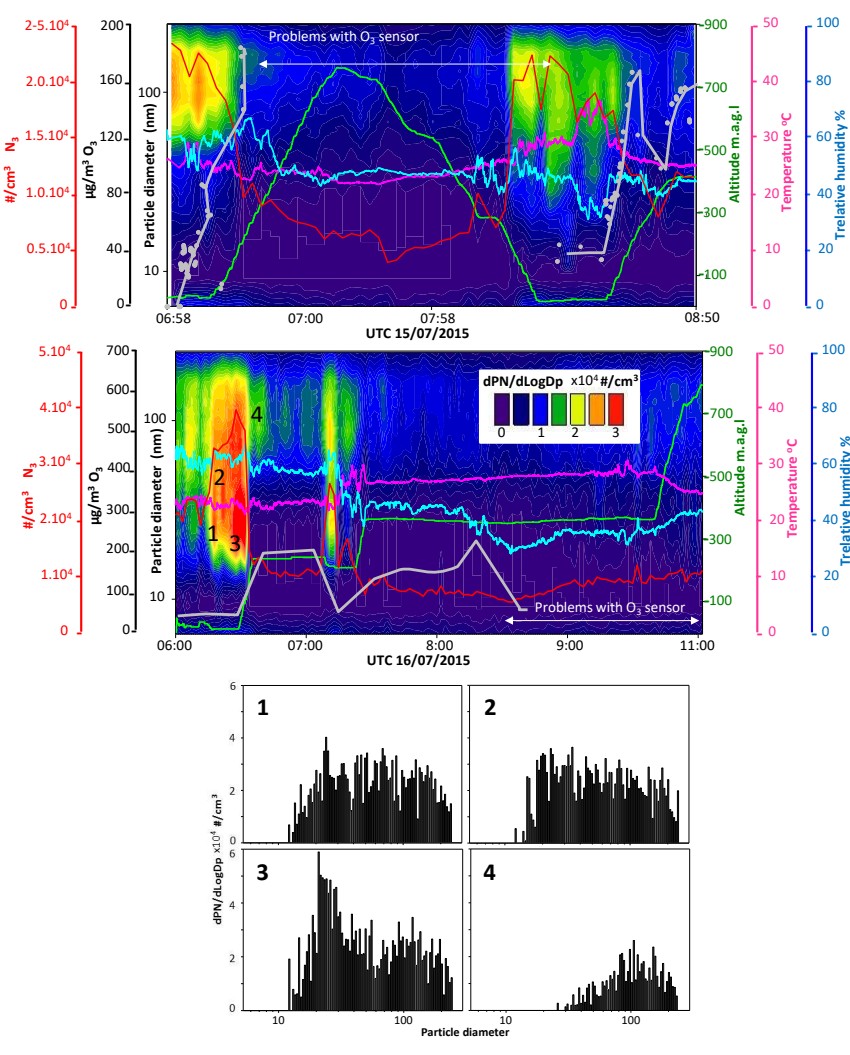

FIGURE 10





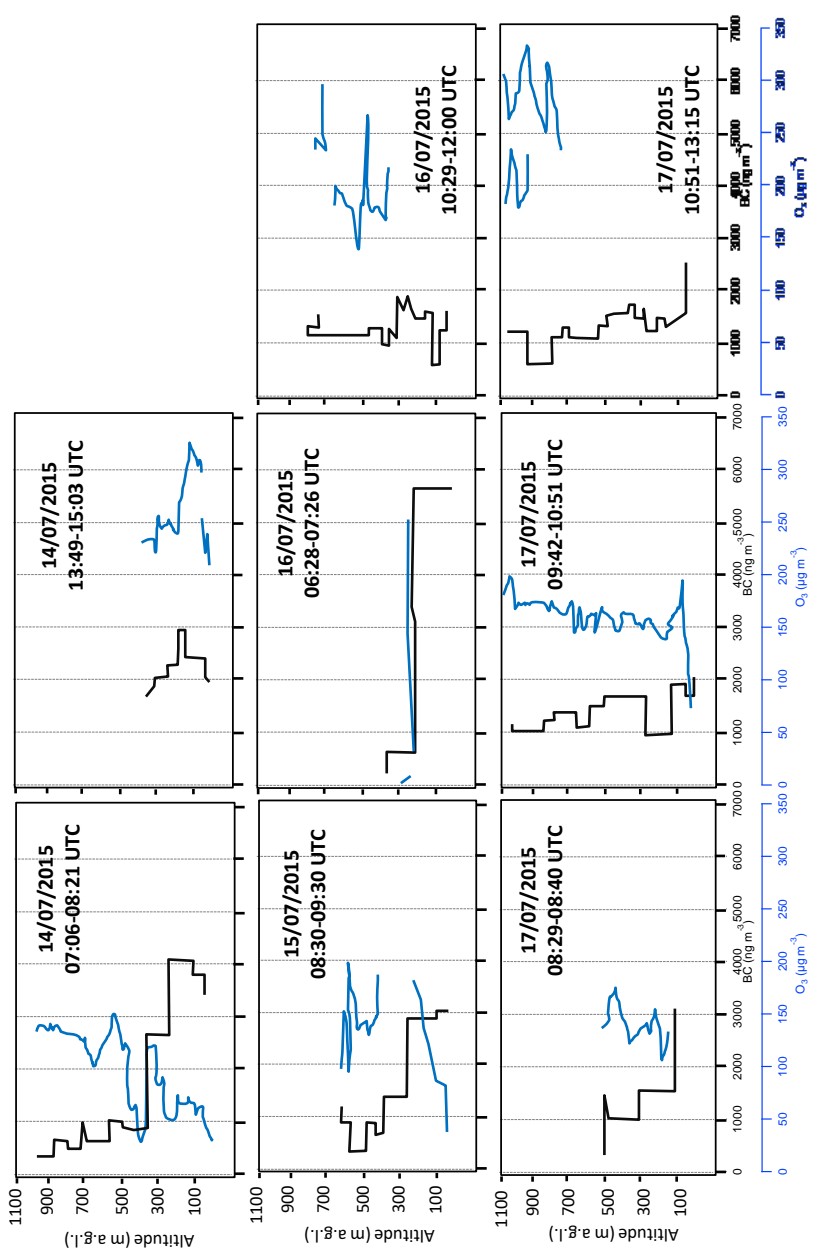

FIGURE 11



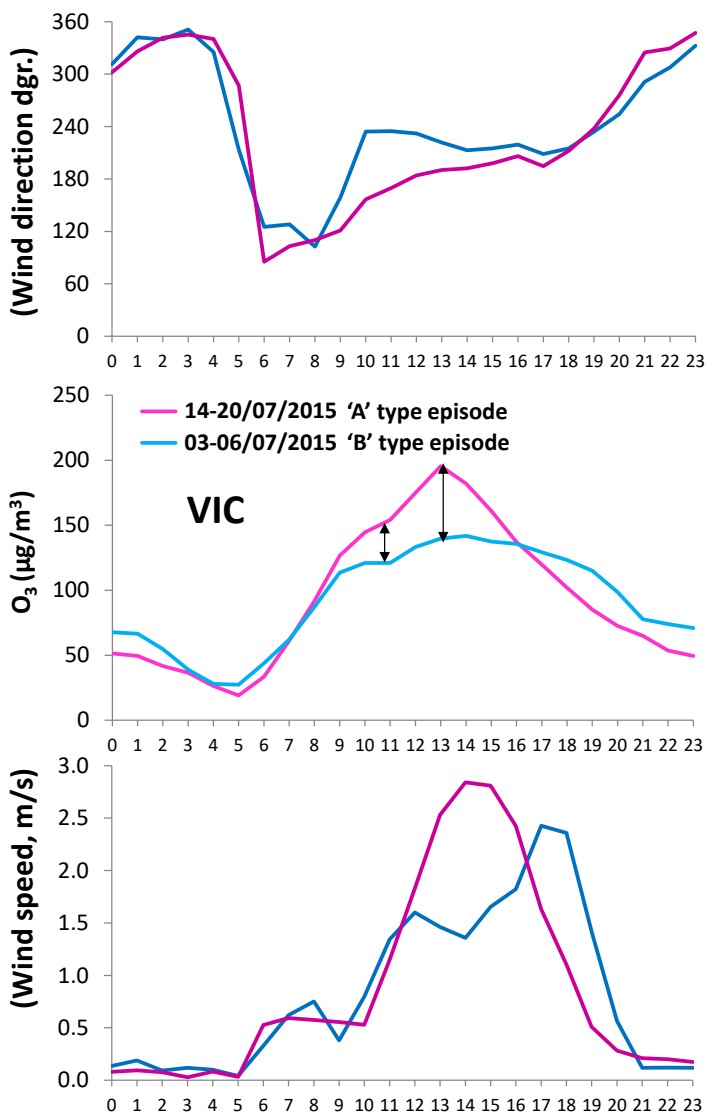

FIGURE 12



FIGURE 13





FIGURE 14





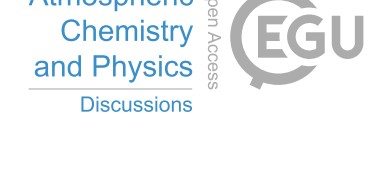

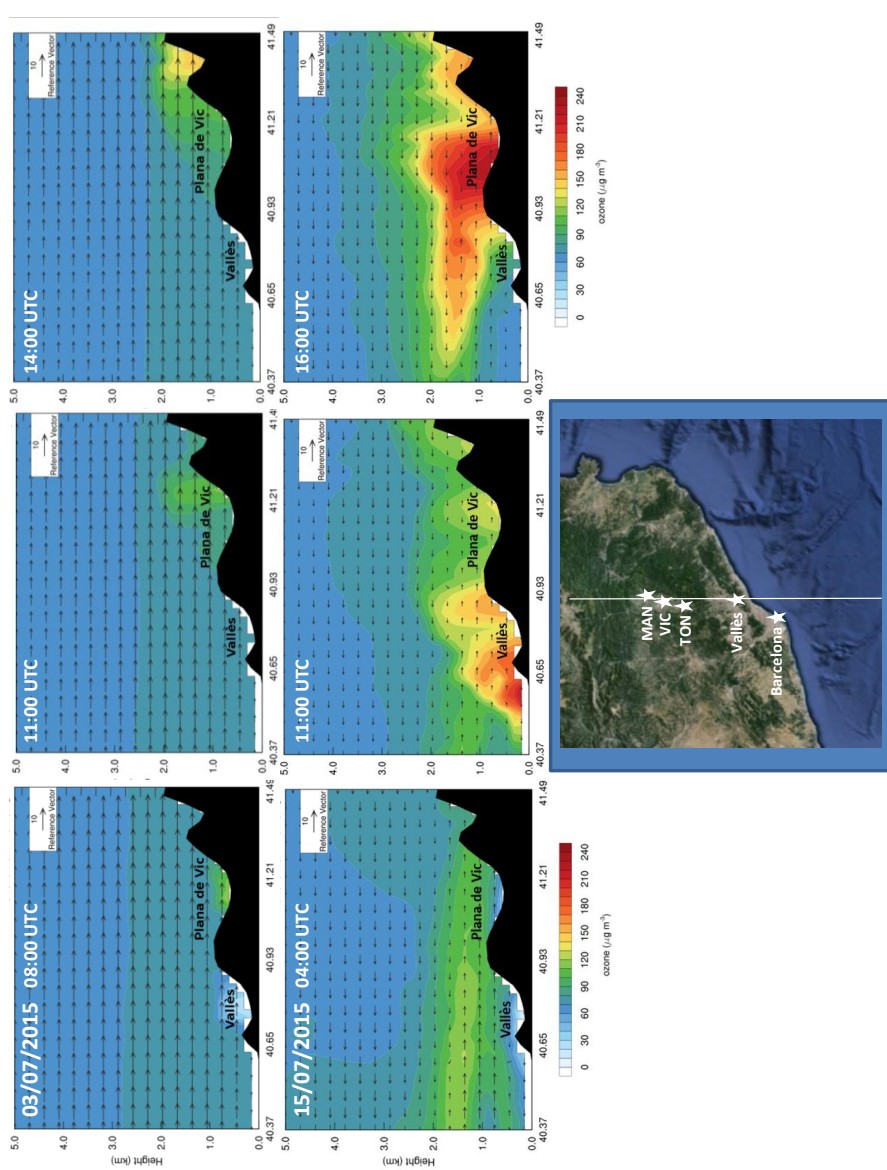

FIGURE 15