# Peer review of "PHENOMENOLOGY OF HIGH OZONE EPISODES IN NE SPAIN"

_Atmospheric Chemistry and Physics, 2016_

## Referee Comment (RC1) · Anonymous Referee #1 · 9 Dec 2016

The manuscript submitted by Querol et al. presents a detailed analysis of the generation of ozone episodes in the Catalunia region (North-Eastern Spain), elucidating key mechanisms yielding to acute ozone episodes in the area. The analysis is carried out exploiting a comprehensive dataset of measurements at ground level and on the vertical profile by means of ballons. The authors were able to identify two types of synoptic patterns associated to high ozone episodes and convincingly describe the underlying processes. For the type associated with highest ozone levels, the authors also suggest that emission reduction in the Barcelona metropolitan area during the days preceding the event, might reduce the risk of having the most severe ozone peaks in the valley to the north of the city. Although specific of the area, the analysis may be taken as a useful example also for other similar areas.

This is a well conceived study reported in a well written manuscript. Publication on

[Figure]

ACP after addressing the minor points listed below is recommended. These are mostly typos and request of clarification at a few points not clear to this reviewer:

1. abstract, l. 24: "vertical measurements". It would be useful to state immediately here that vertical measurements were performed using tethered and non-tethered balloons.

2. abstract, l. 39-40: "At the highest altitudes reached in this study (900-1000 m a.g.l.) ..." this is somewhat in contradiction with the height interval specified above (1500-3000 m a.g.l., see line 29). Please rephrase.

3. abstract, l. 45: "free sounding data". Not immediatly clear what does it mean. I suggest to write "non-tethered balloons" in place of "free".

4. abstract, l. 46-48: unclear paragraph. UFP are said to be low in the lower 100-200 m a.g.l., but nucleation events were detected in the PBL: does it mean that the PBL itself is stratified and nucleation only occurred above 100-200 m? Moreover, the paragraph does not seem to be contain strictly necessary information for an abstract, I would consider removing it at all. Please clarify.

5. abstract, l. 54-57: This paragraph may be removed, it does not add significant and crucial information, as it is written now. Otherwise, please clarify the importance of the statement.

6. l. 78: "(where no exceedances are recommended)": misleading statement. Perhaps rephrase as "(where no specific number of exceedances is recommended)".

7. l. 86: "... to yield secondary aerosols". I would also add "organic nitrates", which may sequester a significant fraction of NOx.

8. l. 108: "In days," replace with "There,".

9. l. 266: "are bivariate polar plots concentrations are ...", add "where" between "plots" and "concentrations"

10. p. 7, subsection "Modelling system for O3": from the description apparently a

continuous run for all the period analysed is carried out (it is mentioned a 24-h spin-up period). In that case, probably grid nudging of WRF was used. Please clarify and eventually specify the variables used in nudging, the nudging coefficients, and if nudging were applied also in the PBL.

11. l. 291: "pressures" should be "pressure".

12. l. 292-293: "... changing the direction at nighttime". Misleading statement. Apparenlty, the meaning is that air masses circulate clockwise during the day and counterclockwise at night. Please clarify.

13. l. 385: "nitrate ... concentrations increased during the evening". This is not true. In Figure S6 nitrate decrease in the evening, while shows a peak in the moring period. Moreover, the effect is attributed to changed "gas/particle partitioning": is this an effect of temperature? Please correct and clarify.

14. l. 420-421: "O3 variations at the coastal BEG are opposed to those at the inland MSC". Not clear what the authors mean with "opposed". The two signals are actually qualitatively correlated. Please clarify.

15. l. 472-477: Here, a quantitative estimate of the non-local contribution to the O3 peak in the Vic Plain is attempted. However, it is not really clear how the authors estimated it. Please clarify, explaining in depth the calculation.

16. l. 488: "... more than 50% of the O3 hourly ...". Again, the authors attempt quantification of different contribution to O3 levels, but do not explain in details the calculations. Please clarify.

17. l. 529: "150 ug/m3". Probably the authors mean "100 ug/m3". Please check.

18. l. 586: " ... more marked in the episode." Probably a "A" is missing at the end of the sentence.

19. l. 586-589: Here the authors refer to previous quantitative estimate of contributions

to O3 levels from specific mechanism. This reinforced the need for clarification, as mentioned above.

20. l. 701-703: Here the authors qualitatively suggest that NOx reduction in BMA should reduce ozone peaks in Vic Plain during type A episodes. They are thus implicitly assuming a NOx-limited ozone regime. Please state ths point explicitly, and possibily support the claim referencing previous studies, if any available.

21. One concluding natural question is: are type A and B the only two situations expected to yield high ozone events in the area? May the authors rule out other types of situation from the analysis of this period only? Please add a comment on that point.

---

## Referee Comment (RC2) · Anonymous Referee #2 · 11 Dec 2016

This study, that is based on a comprehensive set of both ground level and balloon borne measurements of air pollutants as well as model simulations, gives a detailed description of the characteristics of two high ozone episodes observed during the month of July 2015 in the North Eastern part of Spain, where the highest ozone concentrations in the country tyically are found. The authors show that the episode with the highest ozone concentrations is characterized by what they call 'closed circulation' with a high degree of recirculation of air masses due to the sea and land breezes, while during the other episode no major recirculation takes place and horzontal advection over a larger scale plays a more important role. The manuscript builds on a long series of previous studies, mainly carried out by the group of Millan Millan, that have highlighted the influence of the particular orographic and meteorological conditions on air pollution at the Iberian Mediterranean coast and in the western part of the Mediterranean Basin

in general.

The manuscript presents an interesting set of observations and what seems to be a scientifically sound analysis of these; it is well written and generally clear, apart from a few points mentioned in the following. I think that the manuscript would only need minor corrections and recommend that it be published after the authors have addressed the comments and suggestions given below.

As a general comment, I find that while the qualitative description of the contribution of different processes to the episodes is clear, in the cases where a more quantitave evaluation of these contributions to the ozone or Ox-levels is given a more explicit description of the calculations that were performed is needed, as mentioned below.

There seems to be a discrepancy between what is written about the altitude up to which ozone rich layers may influence surface ozone concentrations in different parts of the paper. In the abstract it is stated that surface fumigation takes place "from high O3 reservoir layers located at 1500-3000 m a.g.l. . ..", in accordance with what is written in lines 477-478 and 556-560 but in apparent disagreement with the text in lines 119-120, where the ozone rich layers descending to the surface are said to be located at 1000-1500 m.a.s.l. I realize that there is a difference between 'a.g.l.' and' a.s.l.' but as the layers are descending over the sea it seems that this cannot explain the difference.

In the following I will go through the paper and comment on specific points:

Line 106, 'Seco et al., 2011': The paper by Seco et al. from 2011 is not in the list of references. There is another paper by Seco et al. from 2013, but probably not the one that the authors have in mind because it deals with emissions during wintertime.

Line 135-136, '..which combined with BVOCs emissions, very often cause severe O3 episodes. . .': Is there any study showing that BVOC emissions are dominating VOC reactivity in the Barcelona area? I understand from the paper by Valverde et al. (2016) that VOC emissions from traffic and from the harbour are relatively large in this area.

Lines 145-146, 'SPECIFY SIZE RANGES...': This seems to be a comment left from the internal reviewing process among the authors. I agree with the comment!

Line 336: 'Figure S5' should probably be 'Figure S7'.

Line 414, 'This is due to...': It is possible that the higher ozone levels at the coastal sites may be related to a higher proportion of primary NO2 (due to ship traffic, I suppose), but it remains a hypothesis that this is the main reason for the differences so I think it is mandatory to write 'this may be due to..'.

Line 469: The meaning of the term 'meteorologically influenced patterns' is not completely clear here. I guess that it refers to the impact of long range transport (in contrast to the transport within the region mentioned afterwards), but please change the wording in the text.

Lines 476-481: It is not clear how the 150 micrograms/m3 were calculated. Please give the necessary details.

Lines 493-495: I do not understand the reasoning here: In my understanding not only the inland stations in the Vic plain but also the coastal sites should be subject to fumigation by recirculated strata.

Lines 583-586: The occurrence of layers where ozone and BC are uncorrelated is attributed to recirculation of aged air masses, possibly coming from "local-to-regional sources and more distant over the W-Mediterranean". However in the abstract it is suggested that these layers are "possibly due to a prevailing regional/hemispheric contribution of O3 at those altitudes", i.e. transport at a much larger scale. As this is an important issue and as the abstract should reflect the contents of the paper, I think it would be relevant to discuss this possibility of an impact of long range transport.

Line 591, 'ordered by importance': It is not clear to me how the relative importance of the three processes has been determined.

Technical comments:

The basis for dividing the figures between the main paper and the supplementi is not completely clear to me. I would suggest to put the figures that are most important for the discussion in the main paper. For instance, the maps of the synoptic meteorological situations that lead to the two episodes (type A and type B) are essential for following the discussion in the paper. I would thus suggest to move Figure S5 from the supplement into the main paper and also to replace the present figure with German text by a figure with English text. Also Figure S7 is important for the discussion and thus I find it more natural to have it in the main paper.

Line 180, 'The area is surrounded....': The sentence needs to be rephrased.

---

## Author Comment (AC1) · 12 Dec 2016

**Reply to referees: Phenomenology of the highest ozone episodes in NE Spain" by Querol X. et al.**

Anonymous Referee #1, Received and published: 9 December 2016

We appreciate very much the critical and constructive review made by the referee. As you might see in the revised version we applied all suggestions made by the referee

**REFEREE #1:** The manuscript submitted by Querol et al. presents a detailed analysis of the generation of ozone episodes in the Catalunia region (North-Eastern Spain), elucidating key mechanisms yielding to acute ozone episodes in the area. The analysis is carried out exploiting a comprehensive dataset of measurements at ground level and on the vertical profile by means of balloons. The authors were able to identify two types of synoptic patterns associated to high ozone episodes and convincingly describe the underlying processes. For the type associated with highest ozone levels, the authors also suggest that emission reduction in the Barcelona metropolitan area during the days preceding the event might reduce the risk of having the most severe ozone peaks in the valley to the north of the city. Although specific of the area, the analysis may be taken as a useful example also for other similar areas.

This is a well-conceived study reported in a well written manuscript. Publication on ACP after addressing the minor points listed below is recommended. These are mostly typos and request of clarification at a few points not clear to this reviewer.

**REPLY:** Thanks a lot for your comments and review that greatly helped us to improve the presentation of our results in the paper. As you will see have applied all your suggestions in the revised version. Thanks a lot for your critical and positive review.

**REFEREE #1.1**. Abstract, I. 24: "vertical measurements". It would be useful to state immediately here that vertical measurements were performed using tethered and non-tethered balloons. **REPLY:** Done. Thank you very much.

**REFEREE #1.2.** Abstract, I. 39-40: "At the highest altitudes reached in this study (900-1000 m a.g.l.) ..." this is somewhat in contradiction with the height interval specified above (1500-3000 m a.g.l., see line 29). Please rephrase.

**REPLY:** Done, we clarified in text that in the first case we refer to tethered balloon measurements, and at the higher altitudes was modelling and non-tethered balloon measurements. Thank you very much.

**REFEREE #1.3.** Abstract, I. 45: "free sounding data". Not immediately clear what does it mean. I suggest to write "non-tethered balloons" in place of "free".

REPLY: Done.

**REFEREE #1.4**. abstract, I. 46-48: unclear paragraph. UFP are said to be low in the lower 100-200 m a.g.l., but nucleation events were detected in the PBL: does it mean that the PBL itself is stratified and nucleation only occurred above 100-200 m? Moreover, the paragraph does not seem to contain strictly necessary information for an abstract, I would consider removing it at all. Please clarify.

**REPLY:** Done. We reduced the sentence to: "Relatively low concentrations of ultrafine particles (UFP) during the study, and nucleation episodes were only detected into the boundary layer."

**REFEREE #1.5**. Abstract, I. 54-57: This paragraph may be removed. It does not add significant and crucial information, as it is written now. Otherwise, please clarify the importance of the statement.

REPLY: Done.

**REFEREE #1.6**. I. 78: "(where no exceedances are recommended)": misleading statement. Perhaps rephrase as "(where no specific number of exceedances is recommended)".

**REPLY:** You are right!!! Done.

**REFEREE #1.7.** I. 86: "... to yield secondary aerosols". I would also add "organic nitrates", which may sequester a significant fraction of NOx.

REPLY: Done.

**REFEREE #1.8**. l. 108: "In days," replace with "There,".

REPLY: Done.

**REFEREE #1.9**. I. 266: "are bivariate polar plots concentrations are ...", add "where" between "plots" and "concentrations".

REPLY: Done.

**REFEREE #1.10**. p. 7, subsection "Modelling system for O3": from the description apparently a continuous run for all the period analysed is carried out (it is mentioned a 24-h spinup period). In that case, probably grid nudging of WRF was used. Please clarify and eventually specify the variables used in nudging, the nudging coefficients, and if nudging were applied also in the PBL.

**REFEREE #1.11**. I. 291: "pressures" should be "pressure".

REPLY: Done.

**REFEREE #1.12**. I. 292-293: "... changing the direction at nighttime". Misleading statement. Apparently, the meaning is that air masses circulate clockwise during the day and counterclockwise at night. Please clarify.

**REPLY**: Yes we agree with the misleading of the statement. We re-phrased the paragraph to: "Type A episode: Under "usual summer conditions", with the Azores High located west of Iberia, and a ridge of high pressures extending into southern France, air masses in the Western Mediterranean basin rotate clockwise (anticyclonic) during the day, following the combined sea breezes and upslope flows at eastern Iberia and a simultaneous generalized compensatory sinking is observed in the basin. During nighttime, drainage flows into the sea develop at the coastal strip, subsidence over the basin weakens and the wind over the sea is observed moving southward, transporting the coastal emissions almost parallel to the shoreline (Gangoiti et al., 2001). At the same time, Atlantic

gap winds (through the Ebro and Carcassonne valleys), weaken during daytime due to inland sea breezes and become strengthened during nighttime (Millán et al. 1997; Gangoiti et al., 2001, Gangoiti et al., 2006 and Millán 2014)."

**REFEREE #1.13**. I. 385: "nitrate ... concentrations increased during the evening". This is not true. In Figure S6 nitrate decrease in the evening, while shows a peak in the morning period. Moreover, the effect is attributed to changed "gas/particle partitioning": is this an effect of temperature? Please correct and clarify.".

**REPLY:** Thanks a lot, we corrected and clarified sentence: whereas nitrate (and in a minor proportion ammonium) concentrations increased during the 00:00-08:00 UTC periods as a result of gas/particle partitioning (Figure S6) due to de thermal instability of ammonium nitrate under typical high daytime temperatures (Harrison and Kito, 1990) reached in July 2015 in the study.

Harrison R.M., Kito A.M.N., 1990. Field intercomparison of filter pack and denuder sampling methods for reactive gaseous and particulate pollutants. Atmospheric Environment, 24, 2633–2640.

**REFEREE #1.14**. I. 420-421: " $O_3$  variations at the coastal BEG are opposed to those at the inland MSC". Not clear what the authors mean with "opposed". The two signals are actually qualitatively correlated. Please clarify.

**REPLY:** We clarified in text that the anti-correlation takes place in 01-03, 10-12 and 26/07/2015 and several periods from 14-20/07/2015. You are right that ion several periods these are correlated.

**REFEREE #1.15**. I. 472-477: Here, a quantitative estimate of the non-local contribution to the  $O_3$  peak in the Vic Plain is attempted. However, it is not really clear how the authors estimated it. Please clarify, explaining in depth the calculation.

**REPLY:** We clarified the way it was calculated. Now we stated: "For these exceedances, an hourly contribution of up to 150  $\mu$ g/m3 of Ox (mostly O3) both from fumigation of recirculated return layers (injected at an altitude of 1500-3000 m a.g.l. in the prior day(s)), and from transport and photochemical generation of O3 of the BMA plume, might be estimated based on the differences of the Ox early afternoon maxima recorded at the coastal BMA sites (CTL, PLR) and the ones in the Vic Plain (TON, MON, VIC). Thus, as shown in Figure 5, on 14-18/07/2016 midday maxima recorded at CTL (into BMA) range between 38-62 ppb Ox, on an hourly basis; whereas at TON (in the Vic Plain), these reach 102-115 ppb. Accordingly, differences of 50-73 ppb Ox (close to 100-150  $\mu$ g/m3 Ox) between CTL and TON can be estimated for these days."

**REFEREE #1.16**. l. 488: "... more than 50% of the  $O_3$  hourly ...". Again, the authors attempt quantification of different contribution to  $O_3$  levels, but do not explain in details the calculations. Please clarify.

**REPLY:** We have now already clarified the procedure in the prior paragraph, nonetheless we also clarified this issue here: "As described above, this variation points to the process of  $O_3$  and  $O_x$  formation with a mean  $O_x$  difference between the urban-coastal sites and the Vic Plain hourly maxima of up to 73 ppb  $O_x$  (around 150 µg/m3) for the TON site when subtracted  $O_x$  hourly maxima from CTL (Figure 5), with a maximum average  $O_3$  hourly levels of around 200 µg/m3. These  $O_x$  differences are mostly due to  $O_3$  differences (Figure 6). Accordingly, during these intense  $O_3$  pollution episodes, more than 50% of the  $O_x$  and  $O_3$  hourly maxima concentrations are attributable to....."

**REFEREE #1.17**. I. 529: "150 ug/m3". Probably the authors mean "100 ug/m3". Please check.

**REPLY:** Thanks we checked and modify it!! Now we stated: "On 14/07/2015 07:06-08:21 UTC a well stratified atmosphere (Figure 9) with both thermal and O3 layers is observed, with a general upward increasing trend for O3 from 40  $\mu$ g/m3 at ground level to much higher levels in different strata, such as one reaching 150  $\mu$ g/m3 in strata at 500 and others with 140, 100 or 40  $\mu$ g/m3, such as the ones at 300, 800-1000 or 400 m a.g.l., respectively, reflecting, in addition to stratification of O3 concentrations in altitude, the effect of surface depletion by NO titration and by deposition during the night (see in Figure 9 the progressive O3 depletion from 150  $\mu$ g/m3 at 500 m a.g.l. to 40  $\mu$ g/m3 at surface levels)".

**REFEREE #1.18**. I. 586: " ... more marked in the episode." Probably a "A" is missing at the end of the sentence.

REPLY: Thanks, Yes there was an A missing. We checked and modify it !!

**REFEREE #1.19**. I. 586-589: Here the authors refer to previous quantitative estimate of contributions to  $O_3$  levels from specific mechanism. This reinforced the need for clarification, as mentioned above.

**REPLY:** Thanks, we already explained above in two sections how we calculated.

**REFEREE #1.20**. I. 701-703: Here the authors qualitatively suggest that NOx reduction in BMA should reduce ozone peaks in Vic Plain during type A episodes. They are thus implicitly assuming a NOx-limited ozone regime. Please state this point explicitly, and possibly support the claim referencing previous studies, if any available.

**REPLY:** Sorry for this, we would like to mean 'the reduction of NOx and/or VOCs'. We changed the sentence: "From the perspective of possible precursor abatement strategies, direct mitigation measures at the BMA would have had a minor effect on  $O_3$  concentrations at the Vic Plain area during the type B episode. However, during the type A episode, a reduction of NOx and/or VOCs emissions in the BMA, some days before and during the episode, might have an effect on  $O_3$  concentrations recorded in the Vic Plain." But in any case we also explicitly mentioned in the first version that "Nonetheless, due to the non-linearity of the  $O_3$  generation processes, sensitivity analysis with high resolution modelling is necessary to evaluate the possible effects in terms of absolute concentrations."

**REFEREE #1.21**. One concluding natural question is: are type A and B the only two situations expected to yield high ozone events in the area? May the authors rule out other types of situation from the analysis of this period only? Please add a comment on that point.

**REPLY**: Thanks we have added it: "We are aware that we only analysed the most intense  $O_3$  episodes occurring in July 2015, and that there might be other scenarios, different to type A and B, yielding high  $O_3$  events, such as the transport of aged air masses from other regions of Europe or the transport of the BMA emissions in meteorological scenarios different to those described here. However in a recent study (Querol et al., 2016) we demonstrated with the analysis of the 2000-2015  $O_3$  data series, that the Vic Plain, 40-50 km north of Barcelona) is the area of Spain recording the highest number of annual exceedances of the  $O_3$  information threshold, orders of magnitude higher that the surrounding areas of the axis BMA-Vic Plain-Pre-Pyrenean ranges, thus it is clear that the BMA emissions and the vertical re-circulations caused by the local complex orography have an important role in the occurrence and development of intensive  $O_3$  episodes in the region."

**We added in addition a missing reference:**

Gangoiti, G., L. Alonso, M. Navazo, J. A. García, and M. M. Millán (2006), North African soil dust and European pollution transport to America during the warm season: Hidden links shown by a passive tracer simulation, J. Geophys. Res., 111, D10109, doi: 10.1029/2005JD005941

---

## Author Comment (AC2) · 12 Dec 2016

**Reply to referees: Phenomenology of the highest ozone episodes in NE Spain"**
**by Querol X. et al.**

**Anonymous Referee #2,**

We appreciate very much the critical and constructive review made by the referee. As you might see in the revised version we applied all suggestions made by the referee

**REFEREE #2:** This study, that is based on a comprehensive set of both ground level and balloon borne measurements of air pollutants as well as model simulations, gives a detailed description of the characteristics of two high ozone episodes observed during the month of July 2015 in the North Eastern part of Spain, where the highest ozone concentrations in the country typically are found. The authors show that the episode with the highest ozone concentrations is characterized by what they call 'closed circulation' with a high degree of recirculation of air masses due to the sea and land breezes, while during the other episode no major recirculation takes place and horizontal advection over a larger scale plays a more important role. The manuscript builds on a long series of previous studies, mainly carried out by the group of Millan Millan, that have highlighted the influence of the particular orographic and meteorological conditions on air pollution at the Iberian Mediterranean coast and in the western part of the Mediterranean Basin in general.

The manuscript presents an interesting set of observations and what seems to be a scientifically sound analysis of these; it is well written and generally clear, apart from a few points mentioned in the following. I think that the manuscript would only need minor corrections and recommend that it be published after the authors have addressed the comments and suggestions given below.

**REPLY:** Thanks a lot for your comments and review that greatly helped us to improve the presentation of our results in the paper. As you will see have applied all your suggestions in the revised version. Thanks a lot for your critical and positive review.

**REFEREE #2.1:** As a general comment, I find that while the qualitative description of the contribution of different processes to the episodes is clear, in the cases where a more quantitative evaluation of these contributions to the ozone or Ox-levels is given a more explicit description of the calculations that were performed is needed, as mentioned below.

**REPLY:** Thanks for this comment (that also REFEREE #1 made). See below how we replied to your suggestions of clarifying the way calculations were done.

**REFEREE #2.2:** There seems to be a discrepancy between what is written about the altitude up to which ozone rich layers may influence surface ozone concentrations in different parts of the paper. In the abstract it is stated that surface fumigation takes place "from high $O_3$ reservoir layers located at 1500-3000 m a.g.l: ", in accordance with what is written in lines 477-478 and 556-560 but in apparent disagreement with the text in lines 119-120, where the ozone rich layers descending to the surface are said to be located at 1000-1500 m.a.s.l. I realize that there is a difference between 'a.g.l.' and' a.s.l.' but as the layers are descending over the sea it seems that this cannot explain the difference.

**REPLY:** Sorry for this, there was an error in lines 119-120 and it is 1000-3500 m a.s.l, and not 1000-1500 m.a.s.l. We corrected accordingly in the text.

**REFEREE #2.3:** Line 106, 'Seco et al., 2011': The paper by Seco et al. from 2011 is not in the list of references. There is another paper by Seco et al. from 2013, but probably not the one that the authors have in mind because it deals with emissions during wintertime.

**REPLY:** Yes, sorry for the mistake. The reference as you noticed should be Seco et al 2011. We replaced Seco et al., 2013 by: Seco R., Peñuelas J., Filella I., Llusià J., Molowny-Horas R., Schallhart S., Metzger A., Müller M., Hansel A., 2011. Contrasting winter and summer VOC mixing ratios at a forest site in the Western Mediterranean Basin: the effect of local biogenic emissions. Atmospheric Chemistry and Physics 11, 13161-13179.

**REFEREE #2.4:** Line 135-136, 'which combined with BVOCs emissions, very often cause severe $O_3$ episodes': Is there any study showing that BVOC emissions are dominating VOC reactivity in the Barcelona area? I understand from the paper by Valverde et al. (2016) that VOC emissions from traffic and from the harbour are relatively large in this area.

**REPLY:** Sorry again. Yes, we agree with you that high NOx but also VOCs anthropogenic emissions occur in the BMA. Seco et al., 2011 showed also prevalence of BVOCs in the rural area where high $O_3$ episodes are recorded. In any case we modified the sentence: "High anthropogenic $NO_X$ and VOCs emissions arise both from road (and shipping) traffic and power generation, which combined with BVOCs emissions, very often cause severe $O_3$ episodes in the northern plains and valleys (Toll and Baldasano, 2000; Barros et al., 2003; Gonçalves et al., 2009; Seco et al., 2011; Valverde et al., 2016; Querol et al., 2016)."

**REFEREE #2.5:** Lines 245-246, 'SPECIFY SIZE RANGES: 'This seems to be a comment left from the internal reviewing process among the authors. I agree with the comment!

**REPLY:** Yes, it was a mistake, Thanks a lot. We deleted this message and added the size ranges.

**REFEREE #2.6:** Line 336: 'Figure S5' should probably be 'Figure S7'.

**REPLY:** Yes, thank you. Changed

**REFEREE #2.7:** Line 414, 'This is due to': It is possible that the higher ozone levels at the coastal sites may be related to a higher proportion of primary NO2 (due to ship traffic, I suppose), but it remains a hypothesis that this is the main reason for the differences so I think it is mandatory to write 'this may be due to'

**REPLY**: Totally agree with you. Changed to more open possibilities. "This may be due to…." is now stated.

**REFEREE #2.8:** Line 469: The meaning of the term 'meteorologically influenced patterns' is not completely clear here. I guess that it refers to the impact of long range transport (in contrast to the transport within the region mentioned afterwards), but please change the wording in the text.
**REPLY**: We clarified this important question in 2 paragraphs. The first is in the section of L422-425: "show relatively narrow diurnal variations and multiday episodes, with low or enhanced concentrations, according to meteorological fluctuations (accumulation and air mass renovation cycles of 3 to 12 days cause a wider $O_3$ and $O_X$ concentrations range than the typical daily cycles evidenced in most of the other sites)." The second is in L469: "$O_3$ and $O_X$ concentrations at the regional background site (MSY, 720 m a.s.l., green in Figure 5) depict also the meteorologically influenced patterns (in the sense previously described for BEG and MSC),….."

**REFEREE #2.9:** Lines 476-481: It is not clear how the 150 micrograms/m$^3$ were calculated. Please give the necessary details.

**REPLY:** We clarified the way it was calculated. Now we stated: "For these exceedances, an hourly contribution of up to 150 µg/m$^3$ of $O_X$ (mostly $O_3$) both from fumigation of recirculated return layers (injected at an altitude of 1500-3000 m a.g.l. in the prior day(s)), and from transport and

photochemical generation of $O_3$ of the BMA plume, might be estimated based on the differences of the $O_X$ early afternoon maxima recorded at the coastal BMA sites (CTL, PLR) and the ones in the Vic Plain (TON, MON, VIC). Thus, as shown in Figure 5, on 14-18/07/2016 midday maxima recorded at CTL (into BMA) range between 38-62 ppb $O_X$, on an hourly basis; whereas at TON (in the Vic Plain), these reach 102-115 ppb. Accordingly, differences of 50-73 ppb $O_X$ (close to 100-150 µg/m$^3$ $O_X$) between CTL and TON can be estimated for these days."

Furthermore in L 488 we also clarified this issue: "As described above, this variation points to the process of $O_3$ and $O_X$ formation with a mean $O_X$ difference between the urban-coastal sites and the Vic Plain hourly maxima of up to 73 ppb $O_X$ (around 150 µg/m$^3$) for the TON site when subtracted $O_X$ hourly maxima from CTL (Figure 5), with a maximum average $O_3$ hourly levels of around 200 µg/m$^3$. These $O_X$ differences are mostly due to $O_3$ differences (Figure 6). Accordingly, during these intense $O_3$ pollution episodes, more than 50% of the $O_X$ and $O_3$ hourly maxima concentrations are attributable to….."

**REFEREE #2.10:** Lines 493-495: I do not understand the reasoning here: In my understanding not only the inland stations in the Vic plain but also the coastal sites should be subject to fumigation by recirculated strata.

**REPLY:** In the coastal sites the PBL height is markedly reduced when compared with the inland regions and then the capture of these high altitude $O_3$-rich layers by the PBL growth and the consequent fumigation on the surface is less probable in the coastal areas than in the inland ones. We added this comment in text.

**REFEREE #2.11:** Lines 583-586: The occurrence of layers where ozone and BC are uncorrelated is attributed to recirculation of aged air masses, possibly coming from "local-to-regional sources and more distant over the W-Mediterranean and also from hemispheric transport of air masses". However in the abstract it is suggested that these layers are "possibly due to a prevailing regional/hemispheric contribution of $O_3$ at those altitudes", i.e. transport at a much larger scale. As this is an important issue and as the abstract should reflect the contents of the paper, I think it would be relevant to discuss this possibility of an impact of long range transport.

**REPLY: Thanks a lot for highlighting this important inconsistency. We modified this section to include** "local-to-regional sources, more distant over the W-Mediterranean or even from hemispheric transport of air masses as reported by UNECE (2010)."

**REFEREE #2.12:** Line 591, 'ordered by importance': It is not clear to me how the relative importance of the three processes has been determined.

**REPLY:** We agree also with this. We cannot be completely sure of this order and then we deleted "(ordered by importance)" and we leaved it qualitatively: "…..attributable to fumigation, photochemical production and transport of high $O_3$ air masses, all controlled by insolation."

**REFEREE #2.13:** The basis for dividing the figures between the main paper and the supplementary information is not completely clear to me. I would suggest to put the figures that are most important for the discussion in the main paper. For instance, the maps of the synoptic meteorological situations that lead to the two episodes (type A and type B) are essential for following the discussion in the paper. I would thus suggest to move Figure S5 from the supplement into the main paper and also to replace the present figure with German text by a figure with English text. Also Figure S7 is important for the discussion and thus I find it more natural to have it in the main paper.

**REPLY:** OK, we have moved Figures S5 and S7 to main text and re-numbered all figures accordingly.

**REFEREE #2.14:** Line 180, 'The area is surrounded'. The sentence needs to be rephrased.

**REPLY: Thanks a lot. Yes, we had two missing words. We added them to the text.** "The area is surrounded by mountains and it is affected by thermal inversions during the night."

**We added in addition a missing reference:**

Gangoiti, G., L. Alonso, M. Navazo, J. A. García, and M. M. Millán (2006), North African soil dust and European pollution transport to America during the warm season: Hidden links shown by a passive tracer simulation, J. Geophys. Res., 111, D10109, doi: 10.1029/2005JD005941